



# Robust statistical calibration and characterization of portable low-cost air quality monitoring sensors to quantify real-time $O_3$ and $NO_2$ concentrations in diverse environments

Ravi Sahu[1], Ayush Nagal[2], Kuldeep Kumar Dixit[1], Harshavardhan Unnibhavi[3], Srikanth Mantravadi[4], Srijith Nair[4], Yogesh Simmhan[3], Brijesh Mishra[5], Rajesh Zele[5], Ronak Sutaria[6], Purushottam Kar[2], and Sachchida Nand Tripathi[1]

[1]Department of Civil Engineering, Indian Institute of Technology Kanpur, Kanpur, India
[2]Department of Computer Science and Engineering, Indian Institute of Technology Kanpur, Kanpur, India
[3]Department of Computational and Data Sciences, Indian Institute of Science, Bangalore, India
[4]Department of Electrical Communication Engineering, Indian Institute of Science, Bangalore, India
[5]Department of Electrical Engineering, Indian Institute of Technology, Bombay, India
[6]Centre for Urban Science and Engineering, Indian Institute of Technology, Bombay, India

**Correspondence:** Sachchida Nand Tripath (snt@iitk.ac.in)

**Abstract.** Rising awareness of the health risks posed by elevated levels of ground-level $O_3$ and $NO_2$ have led to an increased demand for affordable and dense spatio-temporal air quality monitoring networks. Low-cost sensors used as a part of Internet of Things (IoT) platforms offer an attractive solution with greater mobility and lower maintenance costs, and can supplement compliance regulatory monitoring stations. These commodity low-cost sensors have reasonably high accuracy but require in-
field calibration to improve precision. In this paper, we report the results of a deployment and calibration study on a network of seven air quality monitoring devices built using the Alphasense $O_3$ (OX-B431) and $NO_2$ (NO2-B43F) electrochemical gas sensors. The sensors were deployed at sites situated within two mega-cities with diverse geographical, meteorological and air quality parameters – Faridabad (Delhi National Capital Region) and Mumbai, India. The deployment was done in two phases over a period of three months. A unique feature of our deployment is a *swap-out* experiment wherein four of these sensors
were relocated to different sites in the two deployment phases. Such a diverse deployment with sensors switching places gives us a unique opportunity to study the effect of seasonal, as well as geographical variations on calibration performance. We perform an extensive study of more than a dozen parametric as well as non-parametric calibration algorithms and find local calibration methods to offer the best performance. We propose a novel local calibration algorithm based on metric-learning that offers, across deployment sites and phases, an average R2 coefficient of 0.873 with respect to reference values for $O_3$
calibration and 0.886 for $NO_2$ calibration. This represents an upto 9% increase in terms of R2 values offered by classical local calibration methods. In particular, our proposed model far outperforms the default calibration models offered by the gas sensor manufacturer. We also offer a critical analysis of the effect of various data preparation and model design choices on calibration performance. The key recommendations emerging out of this study include 1) incorporating ambient relative humidity and temperature as free parameters (or features) into all calibration models, 2) assessing the relative importance
of various features with respect to the calibration task at hand, by using an appropriate feature weighing or metric learning



technique, 3) the use of local (or even hyper-local) calibration techniques such as $k$-NN that seem to offer the best performance in high variability conditions such as those encountered in field deployments, 4) performing temporal smoothing over raw time series data, say by averaging sensor signals over small windows, but being careful to not do so too aggressively, and 5) making all efforts at ensuring that data with enough diversity is demonstrated to the calibration algorithm while training to ensure good generalization. These results offer insights into the strengths and limitations of these sensors, and offer an encouraging opportunity at using them to supplement and densify compliance regulatory monitoring networks.

## 1 Introduction and Related Works

Adverse effects of air pollution have a detrimental impact on the health of human populations as well as the economy (Chowdhury et al., 2018; Landrigan et al., 2018). For instance, high levels of ground-level ozone can cause severe health risks, including but not limited to, difficulty in breathing, increased frequency of asthma attacks, and chronic obstructive pulmonary disease (COPD). The World Health Organization reported (WHO, 2018) that in 2016, 4.2 million premature deaths worldwide could be attributed to outdoor air pollution, 91% of which occurred in low- and middle-income countries where air pollution levels often did not meet its guidelines. Decision-makers require real-time information on air pollution to formulate effective policies which presents a need for monitoring air pollution levels accurately with dense spatio-temporal coverage.

Existing regulatory techniques for assessing urban air quality (AQ) rely on a small network of monitoring stations providing highly precise measurements of the pollutants (Snyder et al., 2013; Malings et al., 2019). In developing countries like India, existing city-level air quality monitoring networks are comprised of a proportionally small number of Continuous Ambient Air Quality Monitoring Stations (CAAQMS). These stations are instrumented with accurate air quality monitoring gas analyzers and Beta-Attenuation Monitors at a commensurately high setup and operating cost. The AQ data offered by a small number of these monitors across a city, however accurate, limit the ability to formulate AQ improvement strategies (Garaga et al., 2018; Fung, 2019). Moreover, CAAQMS with traditional gas analyzers and filter based monitoring facilities are cumbersome and expensive to install, operate and maintain (Sahu et al., 2020). Consequently, real-time actionable data at the citizen level are currently available at very few locations in a city. There is a need for dense air quality monitoring coverage across cities that can complement the limited spatial resolution of existing air pollution maps that are available for citizens (Kumar et al., 2015; Schneider et al., 2017; Zheng et al., 2019). Adequate information on the real-time spatial and temporal distribution of pollutants would allow citizens to make informed decisions, for instance, on their commute (Apte et al., 2017; Rai et al., 2017). However, the overall cost of operating a large number of reference-grade CAAQMS is not practical, especially in developing countries like India.

In recent years, the availability of low-cost AQ monitoring sensors for measuring real-time air pollution concentrations has provided exciting opportunities for finer spatial resolution data (Rai et al., 2017; Baron and Saffell, 2017). The cost of a Federal Reference Method (FRM)-grade monitoring system is around USD 200,000, while that of a low-powered device running commodity AQ sensors is under USD 500 (Jiao et al., 2016; Simmhan et al., 2019). Several low-cost sensors can be installed to complement a few reference monitors for better pollution mapping. In addition, the emergence of cloud computing





and the Internet of Things (IoT) cyber-infrastructure allows building large-scale networks of low-powered AQ monitoring
devices (Baron and Saffell, 2017; Castell et al., 2017; Arroyo et al., 2019). This paves a way for regulatory bodies to use AQ
sensor data to identify patterns and sources of pollution and efficient policy formulation, for scientists to model the interactions
between climate change and pollution accurately (Hagan et al., 2019), and to facilitate the participation of the common public
in citizen science more actively (Gabrys et al., 2016; Commodore et al., 2017; Gillooly et al., 2019; Popoola et al., 2018).

However, the use of low-cost sensor data at a high temporal resolutions presents challenges as available sensors are not
designed to meet rigid performance standards and generate less accurate data than research-grade instruments (Mueller et al.,
2017; Snyder et al., 2013; Miskell et al., 2018). Thus, there is need to evaluate data from real-time sensors and IoT networks
made available by manufacturers of AQ devices, for accuracy and precision (Akasiadis et al., 2019; Williams, 2019).

## 1.1 Challenges in low-cost sensor calibration

Measuring ground-level ozone ($O_3$) and nitrogen dioxide ($NO_2$) accurately using sensors is challenging as they occur at parts
per billion (ppb) micro-levels and intermix with other pollutants (Spinelle et al., 2017). Most commonly available low-cost sen-
sors for these gas-phase compounds are based either on metal oxide (MOx) or electrochemical (EC) technologies (Pang et al.,
2017; Hagan et al., 2019). Field calibration remains one of the major challenges preventing extensive use of these technologies.
Often, sensor calibration is carried out in controlled conditions which differ substantially from real-world conditions.

In addition, these sensors at times have issues of consistency, stability and sensitivity towards environmental conditions,
and cross-sensitivity (Zimmerman et al., 2018; Lewis and Edwards, 2016). For example, $O_3$ electrochemical sensors undergo
redox reactions in the presence of $NO_2$. Further, the constancy of low-cost sensors is recognized to reduce overtime. Moreover,
in electrochemical cells, reagents are spent over time and have a typical lifespan of one to two years (Masson et al., 2015;
Jiao et al., 2016). Thus, there is need for reliable calibration techniques that meet performance metrics required by end-use
applications even at low ambient concentrations (De Vito et al., 2018).

## 1.2 Related Works

Recent works have demonstrated that valid sensor calibration can be achieved by co-locating them with highly accurate
regulatory-grade reference monitors and using various linear and non-linear calibration models (De Vito et al., 2018; Ha-
gan et al., 2019; Morawska et al., 2018). For the specific case of $SO_2$ sensor calibration, Hagan et al. (2019) observed that
simple parametric models such as least squares (LS) regression could extrapolate to wider concentration ranges, at which non-
parametric regression model may fail. However, LS does not correct for the temperature or relative humidity (RH) dependence
of the signal, at which non-parametric models may be more effective.

Since electrochemical sensors are configured such that the responses are diffusion-limited, and the diffusion coefficient could
get affected by varying temperature, Hitchman et al. (1997); Masson et al. (2015) found that at RH exceeding 75% there is
substantial error, possibly due to condensation on the potentiostat electronics. More recently, Simmhan et al. (2019) used non-
parametric approaches such as regression trees along with data aggregated from multiple co-located sensors to demonstrate the
effect of training dataset on calibration performance. Esposito et al. (2016) made use of neural networks and demonstrated good



calibration performance (with mean absolute error < 2 ppb) for calibration of $NO_2$ sensors. However, a similarly impressive performance was not observed for $O_3$ calibration. Moreover, existing works have mostly been tested with localized deployment of a small number of sensor, for instance Cross et al. (2017) who tested two sensor devices, each containing one sensor per

90    pollutant over 4 months with 35% training data.

### 1.3   Our Contributions and the SATVAM initiative

The SATVAM initiative (*Streaming Analytics over Temporal Variables from Air quality Monitoring*) has focused on the development and calibration of low-cost air quality (LCAQ) sensor networks based on highly portable IoT software platforms. These sensors include (see Fig. 1) contain PM2.5 as well as gas sensors. Details on the IoT software platform and SATVAM

95    node cyber infra-structure are available in (Simmhan et al., 2019). The work of Zheng et al. (2019) considered the related problem of dynamic PM2.5 sensor calibration within a sensor network deployed in Delhi, India. The focus of this paper is to build accurate and robust calibration models for the $NO_2$ and $O_3$ gas sensors present in SATVAM devices. Our contributions are summarized below:

1. We report the results of a deployment and calibration study involving 6 sensors deployed at two sites with vastly different

100        meteorological, geographical and air quality parameters, over two phases. A unique feature of our deployment is a *swap-out* experiment wherein 4 of these sensors were relocated to different sites in the two phases (see Sect. 2 for deployment details).

2. The swap-out experiment in particular is crucial in allowing us to investigate the efficacy of calibration models when applied to weather and air quality conditions vastly different from those present during calibration. This is missing from

105        previous works which mostly consider only localized calibration of a couple of models.

3. We present an extensive study of calibration models, both parametric and non-parametric and develop a novel local calibration algorithm based on metric learning that offers both stable (across gases, sites and deployment phases), as well as accurate calibration performance.

4. We present a critical analysis of the effect of data preparation techniques such as volume of data, temporal averaging

110        and data diversity, on calibration performance. This study yields several simple yet crucial take-home messages that significantly boost calibration performance.

### 2   Deployment Setup

Our deployment employed a network of LCAQ sensors as well as reference grade monitors for measuring both $NO_2$ and $O_3$ concentrations, deployed at two sites across two phases. Here we give details of the deployment setup.



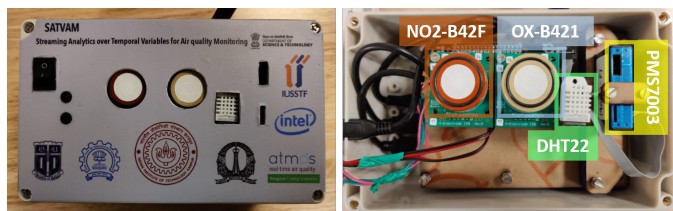

**Figure 1.** Primary components of the *SATVAM* LCAQ (low-cost air-quality) sensor used in our experiments. The SATVAM ensemble consists of a Plantower PMS7003 PM2.5 sensor, Alphasense OX-B431 and NO2-B43F electrochemical sensors, and a DHT22 RH and temperature sensor. Additional components (not shown here) include instrumentation to enable data collection and transmission.

## 2.1 Instrumentation

**Low-cost Sensor Design:** Each *SATVAM* LCAQ device contains two commodity electrochemical gas sensors (Alphasense OX-B421 and NO2-B42F) for measuring $O_3$ (ppb) and $NO_2$ (ppb) levels, a PM sensor (Plantower PMS7003) for measuring PM2.5 (mg m$^{-3}$) levels, and a DHT22 sensor for measuring ambient temperature in °C and RH in percent. Fig. 1 shows the placement of these components. A notable feature of this device is its focus on resource frugality with use of the very low-power ContikiOS platform and 6LoWPAN for providing wireless sensor network communications.

Detailed information on assembling these different components and the cyber-infrastructure required to make a customized sensor node capable of interfacing within an IoT network is available in other works (Simmhan et al., 2019). These works also describe in detail the formation of a highly portable IoT software platform to transmit 6LoWPAN packets at 5 minute intervals containing five time-series data points of the individual sensors, namely $NO_2$, $O_3$, PM2.5 (not presented in this study), temperature and relative humidity (RH). In previous deployments which used only a couple of SATVAM devices, a Raspberry Pi unit was used at each device along with a mesh network to collect and push data to a cloud storage facility. However, for the current deployment that considers a much larger number of devices spread across two cities and seasons, a single border-router edge device was configured at both sites using a Raspberry Pi that acquired data, integrated it, and connected to a cloud facility using a WiFi-link to the respective campus broadband networks. A Microsoft Azure Standard D4s v3 VM was used to host the cloud service with 4 cores, 16 GB RAM and 100 GB SSD storage running an Ubuntu 16.04.1 LTS OS. The Pi edge device was designed to ensure that data acquisition continues even in the event of cloud VM failures.

**Reference Monitors:** At both the deployment sites, $O_3$ and $NO_2$ were measured simultaneously with data available at 1 minute intervals for site D deployments (both Jun and Oct) and 15 minute intervals for site M deployments. $O_3$ and $NO_2$ values were measured at site D using an ultraviolet photometric $O_3$ analyzer (Model 49i $O_3$ analyzer, Thermo Scientific™, USA) and a chemiluminescence oxide of nitrogen (NOx) analyzer (Model 42i NOx analyzer, Thermo Scientific™, USA), respectively. Regular maintenance and multi-point calibration, zero checks, and zero settings of the instruments were carried out following the method described by (Gaur et al., 2014). The lowest detectable limits of reference monitors in measuring $O_3$ and $NO_2$ are 0.5 ppb and 0.40 ppb, respectively, and with a precision of ±0.25 ppb and ±0.2 ppb, respectively. Similarly, the deployments at site M had Teledyne T200 and T400 reference-grade monitors installed. These also have a UV photometric





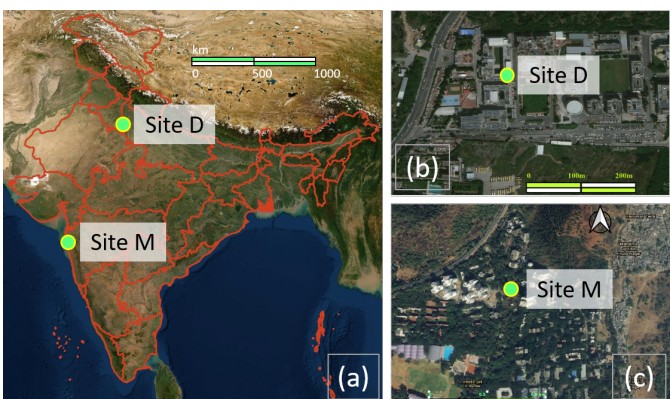

**Figure 2.** A map showing the locations of the deployment sites. Fig. 2(b) and (c) on the right show a local-scale map of the vicinity of the deployment sites – namely Site D at MRIU, **D**elhi NCR (Fig. 2(b)) and Site M at MPCB, **M**umbai (Fig. 2(c)), with the sites themselves pointed out using bright green dots. Fig. 2(a) shows the location of the sites on a map of India. **Credit for Map Sources**: Fig. 2(a) is taken from the NASA Earth Observatory with the outlines of the Indian states in red taken from QGIS3.4 Madeira. Fig. 2(b) and (c) were obtained from, and are, © Google Maps. The green markers for the sites in all figures were added separately.

analyzer to measure $O_3$ levels and use chemiluminescence to measure $NO_2$ concentrations with lowest detectable limits for $O_3$ and $NO_2$ of 0.4 ppb and 0.2 ppb respectively and a precision of $\pm0.2$ ppb and $\pm0.1$ ppb respectively. For every deployment, the reference monitors and the AQ sensors were time-synchronized, with the 1 minute interval data averaged across 15 minute intervals for all site M deployments. The DHT-22 sensor of the SATVAM devices was compared to Vaisala, a reference-grade instrument for temperature and humidity kept alongside the AQ monitors at site D.

**2.2 Deployment Sites**

SATVAM LCAQ sensor sensor deployment and collocation with reference monitors was carried out at two sites. Fig. 2 presents the geographical locations of these two sites.

1. **Site D**: located within the **D**elhi National Capital Region (NCR) of India at the Manav Rachna International Institute of Research and Studies, Sector 43, Faridabad (28.45°N, 77.28°E, 209 m above mean sea level).

2. **Site M** (in Mumbai): located within the city of **M**umbai at the Maharashtra Pollution Control Board within the university campus of IIT Bombay (19.13°N, 72.91°E, and 50 m above mean sea level).

 **About Site D**: According to Greenpeace India (Times, 2018) and the Niti Aayog, Govt. of India (Aggarwal, 2018), Faridabad was the second most polluted city in India in 2018. Surrounded by the Aravalli Hills, this is a rapidly growing city and a leading industrial center suffering from heavy air pollution that mask the city and its neighborhoods routinely during the fall and winter

seasons. The study site is 5 km away from Delhi and near Delhi-Surajkund Highway. It falls in the Indo-Gangetic Plain, which registered critical levels of ambient air pollution attributable to a combination of multiple ambient sources, the use of biomass





| | Sensors | | | | | |
|---|---|---|---|---|---|---|
| | DD1 | **DM2** | DD3 | MM5 | **MD6** | **MD7** |
| Jun | D | D | D | M | M | M |
| Oct | D | M | D | M | D | D |

| Jun deployment | Oct deployment |
|---|---|
| **Site D** DD1, **DM2** DD3 | **Site D** DD1, DD3 **MD6, MD7** |
| **Site M** MM5, **MD6** **MD7** | **Site M** **DM2**, MM5 |

**Figure 3.** A schematic showing the deployment of the LCAQ sensors across site D and site M during the two deployments. The sensors subjected to the *swap-out* experiment are presented in bold. The outlines of the Indian states in red was taken from QGIS3.4 Madeira with other highlights (e.g. for oceans) and markers being added separately.

and coal for household cooking and heating needs, and the stubble or agricultural residue burning (Chowdhury et al., 2019). The deployment site is affected by vehicular traffic which are likely a dominant source of precursors to $O_3$ formation ($NO_2$ and volatile organic compounds) and of nitric oxide that reacts with $O_3$ to form the pollutant $NO_2$. The reference monitors were deployed in a laboratory on the first floor of the building with the low-cost AQ monitoring sensors next to its inlets.

**About Site M**: This site presents relatively lower pollution levels as it is situated within the IIT Bombay campus between the Vihar and Powai lakes, and it is adjacent to the Sanjay Gandhi National park. Less that 5 km to its west side passes the Thane creek (an inlet in the shoreline of the Arabian Sea) that isolates the city of Mumbai from the Indian mainland while the Arabian sea is at around 10 km to its west. The reference monitors were deployed on the rooftop of the building with the low-cost AQ monitoring sensors next to its inlets. All AQ monitoring devices were in a Stevenson box to avoid damage to sensors.

Due to ever-increasing economic and industrial activities across the city, a progressive worsening of ambient air pollution is nearly inevitable at both study sites. We considered these two polluted sites situated within the Delhi-NCR and Mumbai to cover a broader range of pollutant concentrations and weather patterns, so as to be able to test the reliability of low-cost sensor networks in measuring $O_3$ and $NO_2$ levels. It is notable that the two chosen sites present different geographical settings as well as different air pollution levels with site D of particular interest in presenting significantly higher minimum $O_3$ levels than site M, illustrating the influence of the geographical variability over the selected region.





## 2.3 Deployment Details

A total of four field co-location deployments, two each at sites D and M, were evaluated to characterize the calibration of the low-cost sensors during two seasons of 2019. The two field deployments at site D were carried out from 27th Jun–6th Aug

2019 (7 weeks) and 4th Oct–27th Oct 2019 (3 weeks). The two field deployments at site M, on the other hand, were carried out from 22nd Jun–21st Aug 2019 (10 weeks), and 4th Oct–27th Oct 2019 (3 weeks) respectively. For sake of convenience, we will refer to both deployments that commenced in the month of June 2019 (resp. October 2019) as *Jun* (resp. *Oct*) deployments even though the dates of both Jun deployments do not exactly coincide.

     A total of six low-cost SATVAM LCAQ sensors were deployed at these two sites. We assign these sensors a unique numerical

identifier and a name that clearly depicts its deployment pattern. The name of a sensor is of the form XYn where X (resp Y) indicates the site at which the sensor was deployed during the Jun (resp Oct) deployment and n denotes its unique numerical identifier. The seven sensors are thus named DD1, DM2, DD3, MM5, MD6, and MD7. Fig. 3 outlines the deployment patterns.

## 2.4 Swap-out Experiment

As Fig. 3 indicates, two of the sensors from each of the sites were exchanged or swapped out to the other city across the two

deployments. DM2 was shifted from Delhi to Mumbai and MD6 and MD7 were shifted from Mumbai to Delhi for the Oct deployment.

## 3 Data Analysis Setup

**Testbench:** All experiments were conducted on a commodity laptop with an Intel Core i7 CPU with 2.70GHz frequency, 8GB of system memory and running an Ubuntu 18.04.4 LTS operating system. Standard off-the-shelf machine learning and

statistical analysis packages such as numpy, sklearn, scipy and metric–learn were used to implement the calibration algorithms.

### 3.1 Raw Datasets and Features

The six sensors across the Jun and Oct deployments, gave us a total of 12 datasets. We refer to each dataset by mentioning the sensor name and the deployment name. For example, the dataset DM2(Oct) contains data from the October deployment (at site M) of the sensor DM2. Each dataset is represented as a collection of eight time series for which each time stamp is represented

as an 8-tuple (O3, NO2, RH, T, no2op1, no2op2, oxop1, oxop2) giving us, respectively, the reference values for $O_3$ and $NO_2$ (in ppb), relative humidity (in %) and temperature (in °C) values at each time stamp, in addition to voltage readings (in mV) from the two electrodes present in each of the two gas sensors. These readings are named no2op1, no2op2, oxop1, and oxop2 and they represent working (no2op1 and oxop1) and auxiliary (no2op2 and oxop2) electrode potentials for these sensors.





**Table 1.** Samples of the raw data collected from the DM2(Jun) and MM5(Oct) datasets. The last column indicates whether data from that time-stamp was used in the analysis or not. Note that DM2(Jun) data, coming from site D, has samples at 1 minute intervals whereas MM5(Oct) data, coming from site M, has samples at 15 minute intervals. The raw voltage values (no2op1, no2op2, oxop1, oxop2) offered by the LCAQ sensor are always integer valued, as indicated in the DM2(Jun) data. However, for site M deployments, due to averaging, the effective voltage values used in the dataset may be fractional, as indicated in the MM5(Oct) data. The symbol × indicates missing values.

| | DM2(Jun) | | | | | | | | | | |
|---|---|---|---|---|---|---|---|---|---|---|---|
| Time-stamp | O3 | NO2 | T | RH | no2op1 | no2op2 | oxop1 | oxop2 | no2diff | oxdiff | Valid? |
| 29-06 04:21 | 19.82 | 20.49 | 32.7 | 54.6 | 212 | 231 | 242 | 209 | -19 | 33 | Yes |
| 29-06 04:22 | 21.89 | 20.56 | 32.7 | 54.6 | 212 | 231 | 243 | 210 | -19 | 33 | Yes |
| 29-06 04:23 | 22.71 | 18.17 | × | × | × | × | × | × | × | × | No |
| 29-06 04:24 | 24.82 | 14.60 | 32.5 | 53.3 | × | × | × | × | × | × | No |
| | MM5(Oct) | | | | | | | | | | |
| Time-stamp | O3 | NO2 | T | RH | no2op1 | no2op2 | oxop1 | oxop2 | no2diff | oxdiff | Valid? |
| 17-10 05:45 | 7.17 | 37.62 | 26.12 | 99.9 | 119.27 | 152.93 | 128 | 133.4 | -33.67 | -5.4 | Yes |
| 17-10 06:00 | 8.7 | 34.11 | 26.14 | 99.9 | 122.93 | 155.53 | 131.87 | 136.47 | -32.6 | -4.6 | Yes |
| 17-10 06:15 | × | × | 26.25 | 99.9 | 121.67 | 154.13 | 129.2 | 134.6 | -32.47 | -5.4 | No |
| 17-10 06:30 | 10.86 | 30.95 | 26.16 | 99.9 | 119.33 | 151.4 | 127.27 | 131.67 | -32.07 | -4.4 | Yes |

## 3.2 Data Cleanup

Time-stamps from each of the LCAQ sensors were aligned to those from the reference monitors. We considered only those datapoint that were temporally aligned. For several time-stamps, we found that either the sensor or reference monitors presented with one or more missing or spurious values (see Table 1 for examples). Spurious values included a temperature reading of > 50 °C or < 1 °C, an RH level of > 100 % or < 1 %, a reference value for $O_3$ or $NO_2$ of > 200 ppb or < 1 ppb, or voltage readings from the four sensors at values either > 400 mV or < 1 mV. These errors are possibly due to electronic noise in the 205 devices. All time-stamps with even one spurious or missing value were considered invalid and removed. Across all 12 datasets, an average of 52% of the time-stamps were removed as a result.

For site D deployments, both the LCAQ sensor as well as the reference monitor data was available at 1 minute intervals. However for site M deployments, whereas the LCAQ sensors continued to provide data at 1 minute intervals, the reference monitors at that site were set to provide data at 15 minute intervals. To align the two time series, LCAQ sensor data was 210 averaged over 15 minute intervals.

The 3 datasets from Jun (resp. 4 from Oct) deployments at site D offered an average of 33753 (resp. 9548) valid time-stamps. The 3 datasets from Jun (resp. 2 from Oct) deployments in site M offered an average of 2462 (resp. 1062) valid time-stamps. As expected, site D deployments offered more valid time-stamps than site M deployments in any season given that the former enjoyed 1 minute interval data whereas the latter deployments had data at 15 minute intervals. We also note that for both sites, 215 more data is available for the Jun deployment (that lasted longer) than the Oct deployment.





## 3.3 Data Augmentation and Derived Dataset Creation

For each of the 12 datasets, apart from the six data features provided by the LCAQ sensors, namely RH and T values and sensor voltage values (no2op1, no2op2, oxop1, oxop2), we included two derived features, calculated as shown below

no2diff = no2op1 − no2op2

oxdiff = oxop1 − oxop2

We found that having these derived features, albeit simple linear combinations of raw features, offered our calibration models a predictive advantage. The *augmented* datasets created this way represented each time-stamp as a vector of 8 feature values (RH, T, no2op1, no2op2, oxop1, oxop2, no2diff, oxdiff), apart from the reference values of $O_3$ and $NO_2$.

In order to study the effect of data frequency (how frequently do we record data e.g. 1 minute, 5 minute), data volume
(total number of time-stamps used for training), and data diversity (data collected across seasons or cities) on the calibration performance, we created several *derived* datasets as well. All these datasets contained the augmented features.

1. **Temporally Averaged Datasets**: We took the two datasets DD1(Jun) and DM2(Jun) and created four datasets out of each of them by averaging the sensor and reference monitor values at 5 minute, 15 minute, 30 minute and 60 minute intervals. These datasets were named by affixing the averaging interval size to the dataset name, for example DD1(Jun)-
AVG5 for the dataset created out of DD1(Jun) with 5 minute averaging, DM2(Jun)-AVG30 for the dataset created out of DM2(Jun) with 30 minute averaging, etc.

2. **Sub-sampled Datasets**: To view the effect of having less training data on calibration performance, we created *sub-sampled* versions of both these datasets by sampling a random set of 2500 time-stamps to get the datasets DD1(Jun)-SMALL and DM2(Jun)-SMALL.

3. **Aggregated Datasets**: Next, we created new datasets by clubbing together data for a sensor across the two deployments. This was done to the data from the sensors DD1, MM5, DM2 and MD6. For example, if we consider the sensor DD1, then the datasets DD1(Jun) and DD1(Oct) were combined to create the dataset DD1(Jun-Oct). This was done in order to study the effect of offering data to the calibration models that is more diverse in terms of location (since DM2 and MD6 moved across sites) and season (Jun vs Oct).

### 3.3.1 Train–Test Splits

To create training and test data from each dataset (whether original or derived), we randomly split each dataset in a 70:30 ratio to obtain a train-test split. 10 such splits were independently generated for each dataset. All calibration algorithms were offered the same train-test splits. For algorithms that required hyperparameter tuning, a randomly chosen set of 30% of the training samples in that split were used as a held out validation set. All features were normalized to improve the conditioning of the
calibration problems. This was done by calculating the mean and standard deviation for each of the 8 features on the training portion of a split, and then mean centering and dividing by the standard deviation all time-stamps in both training and testing





portion of that split. An exception was made for the Alphasense calibration models, which required raw voltage values. Also, the reference values were never normalized in any way.

### 3.3.2  Error Metrics and Statistical Hypothesis Testing

**Error Metrics**: calibration performance was measured using four popular metrics, mean averaged error (MAE), mean absolute percentage error (MAPE), root mean squared error (RMSE), and the coefficient of determination ($R^2$) (see below). Here $n$ denotes the number of test points for a given dataset and split thereof, the variable $t$ runs over all time-stamps in the testing set, $y^t$ denotes the reference value (either $O_3$ or $NO_2$) at the $t$-th time-stamp, $\hat{y}^t$ denotes the corresponding value predicted by the calibration model, and $\bar{y}$ denotes the mean reference value i.e. $\bar{y} = \frac{1}{n}\sum_{t=1}^{n} y^t$.

$$\text{MAE} = \frac{1}{n}\sum_{t=1}^{n} |y^t - \hat{y}^t|$$

$$\text{MAPE} = \frac{1}{n}\sum_{t=1}^{n} \frac{|y^t - \hat{y}^t|}{y^t} \times 100\%$$

$$\text{RMSE} = \sqrt{\frac{1}{n}\sum_{t=1}^{n} (y^t - \hat{y}^t)^2}$$

$$\text{R}^2 = 1 - \frac{\sum_{t=1}^{n} (y^t - \hat{y}^t)^2}{\sum_{t=1}^{n} (y^t - \bar{y})^2}$$

**Statistical Hypothesis Tests**: in order to compare the performance of different calibration algorithms on a given dataset (e.g., to find out the best performing algorithm), or compare the performance of the same algorithm on different datasets (e.g., to find out the effect of data characteristics on calibration performance), we performed paired and unpaired two-sample tests, respectively. Our null hypothesis in all such tests proposed that the absolute errors offered by the two algorithms on the same dataset (in case of a paired test) or the same algorithm across different datasets (in case of an unpaired test) were sampled from the same distribution. The test was applied and if the null hypothesis was rejected with sufficient confidence (an $\alpha$ value of 0.05 was used as the standard to reject the null hypotheses), then a winner was simultaneously identified.

Although the Student's t-test is most popularly used in such situations, it essentially assumes that the underlying distributions are normal. However, an application of the Shapiro-Wilk test (Shapiro and Wilk, 1965) rejected the null hypotheses of the errors being normally distributed with high confidence. As a result, we chose the non-parametric Wilcoxon signed-rank test (Wilcoxon, 1945) when comparing two algorithms on the same dataset, and its unpaired variant, the Mann-Whitney $U$-test (Mann and Whitney, 1947) for comparing the same algorithm on two different datasets. These tests do not make any assumption on the underlying distribution of the errors and are well-suited for our data.

## 4  Calibration Models

Our study used a large number of both parametric, and non-parametric calibration techniques. Since several of these techniques are standard, we describe them in the *Supporting Information* document supplied with this paper. In particular, the Supporting





Information document describes several parametric calibration algorithms including the Alphasense models supplied by the manufacturers of the gas sensors, linear models based on least-squares and sparse recovery, as well as several non-parametric calibration algorithms such as regression trees, kernel-ridge regression, and the Nystroem method. We describe here in the main paper, only those baseline calibration models upon which our proposed technique is developed.

**Notation**: For every time-stamp $t$, the vector $\boldsymbol{x}^t \in \mathbb{R}^8$ denotes the 8-dimensional vector of signals recorded by the LCAQ sensors for that time-stamp, namely (RH, T, no2op1, no2op2, oxop1, oxop2, no2diff, oxdiff), while the vector $\boldsymbol{y}^t \in \mathbb{R}^2$ will denote the 2-tuple of the reference values of $O_3$ and $NO_2$ for that time step. However, this notation is unnecessarily cumbersome since we will build separate calibration models for $O_3$ and $NO_2$. Thus, to simplify the notation, we will instead use $y^t \in \mathbb{R}$ to denote the reference value of the gas being considered (either $O_3$ or $NO_2$). The goal of calibration will then be to learn a real valued function $f : \mathbb{R}^8 \to \mathbb{R}$ such that $f(\boldsymbol{x}^t) \approx y^t$ for all time-stamps $t$ (the exact error being measured using metrics such as MAE, MAPE, etc described in Sect. 3.3.2). Thus, we will learn two functions, say $f_{NO_2}$ and $f_{O_3}$ to calibrate for $NO_2$ and $O_3$ concentrations respectively. Since several of our calibration algorithms will involve the use of some statistical estimation or machine learning algorithm, we will let $N$ (resp. $n$) denote the number of training (resp. testing) points for a given dataset and split thereof. Thus, we will let $\{(\boldsymbol{x}^t, y^t)\}_{t=1}^N$ denote the training set for that dataset and split with $\boldsymbol{x}^t \in \mathbb{R}^8$ and $y^t \in \mathbb{R}$.

## 290 4.1 $k-$NN Regression Variants

The $k$-nearest neighbor algorithm is a *local* proximity-based learning algorithm that makes predictions on test samples based on which are the training samples that most resemble the test sample. Resemblance is usually calculated using a metric such as the Euclidean metric. We implement several $k$-nearest neighbor variants. Algorithm 1 gives pseudo code for these variants.

$k$-**NN with Euclidean Distance (KNN):** The vanilla $k$-nearest algorithm (KNN) predicts on a test sample, the average reference value in the $k$ nearest training samples. The neighborhood size $k$ was tuned over the fine grid $[2, 4, 6, 8, 10, 15, 20]$ using held-out validation. Standard implementation of kd-trees were used to accelerate the process of discovering the nearest neighbors for a test sample.

**Distance weighted $k$-NN (KNN-D):** We also implemented a *distance-weighted* version of this algorithm wherein closest neighbors for particular test sample are weighted according to their Euclidean distance to the test point with closer points getting more weightage. We found this to favorably improve our accuracy.

## 4.2 Kernel Regression Variants

In statistics and machine learning, the notion of a *kernel* refers to a function that assigns a similarity value to two vectors (Mur-
phy, 2012). Thus, a kernel is of the form $K : \mathbb{R}^8 \times \mathbb{R}^8 \to \mathbb{R}$ which, when given two vectors $\boldsymbol{x}^1, \boldsymbol{x}^2 \in \mathbb{R}^8$, assigns a value $K(\boldsymbol{x}^1, \boldsymbol{x}^2) \in \mathbb{R}$ denoting how similar are these vectors. A popularly used kernel is the *Gaussian* kernel (aka the *RBF* kernel) that calculates this similarity as $K(\boldsymbol{x}^1, \boldsymbol{x}^2) = \exp(-\gamma \cdot \|\boldsymbol{x}^1 - \boldsymbol{x}^2\|_2^2)$ where $\|\cdot\|_2$ denotes the Euclidean norm and $\gamma$ is a *band-*





*width* parameter that controls the scale at which similarity values go down. The Nadaraya-Watson estimator and kernel ridge regression are two popular forms of kernel regression algorithms. A closely related cousin is Gaussian-process regression.

Below we describe the Nadaraya-Watson estimator as it is useful in the developement of our proposed technique. Kernel ridge regression is described in the Supporting Information document.

**Nadaraya-Watson (NW):** Given a training set $\{(\boldsymbol{x}^t, y^t)\}_{t=1}^N$, the NW estimator (Nadaraya, 1964; Watson, 1964) makes a prediction on a new (testing) data point $\boldsymbol{x} \in \mathbb{R}^8$ as follows

$$f^{\mathrm{NW}}(\boldsymbol{x}) = \frac{\sum_{t=1}^N y^t \cdot K(\boldsymbol{x}^t, \boldsymbol{x})}{\sum_{t=1}^N K(\boldsymbol{x}^t, \boldsymbol{x})}$$

The intent of this estimator is clear – the final prediction is a weighted sum of reference values $y^t$ in the training set with the weight of a training sample $t \in [N]$ being proportional to $K(\boldsymbol{x}^t, \boldsymbol{x})$ i.e. how similar is that training sample to the test sample. Notice also the similarity between NW and KNN-D in the way they make predictions. NW almost behaves like a "smoothed" version of KNN-D by performing weighing using kernel values instead of inverse Euclidean distances and considering all

training samples instead of just the neighbors. This observation will be useful later.

## 5    Proposed Calibration Model: $k$-NN variants with a learnt metric

Below we propose a novel application the *metric learning* technique to build non-parametric calibration models that offer superior performance compared to other models.

### 5.1    Metric Learning

As Sect. 6 will show, the inclusion of RH and T as additional features benefits calibration performance. However, it is unclear how much importance should these features receive as opposed to the other features that are based on voltage readings (e.g. no2op1, oxop2, no2diff etc). This is particularly true of $k$-NN and kernel regression, both of which find neighbors or calculate kernel values by relying on the Euclidean distance which assigns equal importance to all 8 features.

It is well established that $k$-NN style algorithms stand to gain if used with a customized metric instead of the generic

Euclidean metric (Weinberger and Saul, 2009). It is most popular to replace the Euclidean metric with a learnt *Mahalanobis metric*. This metric is characterized by a positive semi-definite matrix $\boldsymbol{\Sigma} \in \mathbb{R}^{8 \times 8}$ and calculates the distance between any two points as follows

$$d^{\mathrm{Maha}}(\boldsymbol{x}^1, \boldsymbol{x}^2; \boldsymbol{\Sigma}) = \sqrt{(\boldsymbol{x}^1 - \boldsymbol{x}^2)^\top \boldsymbol{\Sigma} (\boldsymbol{x}^1 - \boldsymbol{x}^2)}$$

Note that the Mahalanobis metric recovers the Euclidean metric when $\boldsymbol{\Sigma} = I_8$ is the identity matrix. Now, whereas metric

learning for $k$-NN is very popular for classification problems, it is uncommon for regression problems. This is partly because of technical problems posed by regression problems which lack of a small number of "classes".



---

**Algorithm 1** Variants of $k$-NN based calibration

---

**Require:** feature vector for a test sample $\tilde{\boldsymbol{x}}$, training samples $\{(\boldsymbol{x}^t, y^t)\}_{t=1}^N$, neighborhood size $k$, weighing rule, metric

**Ensure:** a prediction $\hat{y}$ for the test sample using one of the KNN, KNN-D, KNN(ML) or KNN-D(ML) calibration model depending on the weighing rule and metric arguments

    **if** metric == Euclidean **then**

        $\boldsymbol{\Sigma} \leftarrow I_8$                                                             `{The 8×8 Identity matrix}`

    **else if** metric == learnt **then**

        $\boldsymbol{\Sigma} \leftarrow$ use training samples to learn a Mahalanobis metric using the technique from (Weinberger and Tesauro, 2007)

    **end if**

    Find the $k$ training samples (say $i^1, \ldots, i^k$) that are closest to $\tilde{\boldsymbol{x}}$ in terms of the learnt Mahalanobis distance $d^{\text{Maha}}(\cdot, \cdot; \boldsymbol{\Sigma})$

    **if** weighing rule == uniform **then**

        $\hat{y} = \frac{1}{k} \sum_{l=1}^{k} y^{t_l}$

    **else if** weighing rule == distance weighted **then**

        For all $l = 1 \ldots k$, let $\alpha^l = (d^{\text{Maha}}(\boldsymbol{x}, \boldsymbol{x}^{t_l}; \boldsymbol{\Sigma}))^{-1}$

        $\hat{y} = \frac{\sum_{l=1}^{k} \alpha^l \cdot y^{t_l}}{\sum_{l=1}^{k} \alpha^l}$

    **end if**

    **return** $\hat{y}$

---

To overcome this problem, we recall our earlier observation that the NW algorithm almost behaves like a smoothed version of the KNN-D algorithm. Given that there does exist a technique (Weinberger and Tesauro, 2007) to learn a Mahalanobis metric for use with the NW algorithm, we adopt a *two-stage* algorithm that first learns a metric suited for the NW estimator and then

using it with the KNN and KNN-D algorithms. The method proposed by Weinberger and Tesauro (2007) learns the metric by attempting to minimize the leave-one-out RMSE over the training samples.

**Metric learning with Nadaraya-Watson (NW(ML)) and $k$-NN algorithms (KNN(ML), KNN-D(ML)):** We call the variants of NW, KNN and KNN-D when used with a learnt metric, respectively NW(ML), KNN(ML) and KNN-D(ML). The

modification required to execute NW(ML) with a learnt metric is straightforward – we simply start using an alternate kernel given by

$$K^{\text{Maha}}(\boldsymbol{x}^1, \boldsymbol{x}^2; \boldsymbol{\Sigma}) = \exp(-(d^{\text{Maha}}(\boldsymbol{x}^1, \boldsymbol{x}^2; \boldsymbol{\Sigma}))^2)$$

We note that this alternate kernel does not require an explicit bandwidth parameter since any such parameter can be absorbed into the matrix $\boldsymbol{\Sigma}$ itself. Algorithm 1 details pseudo-code for KNN(ML) and KNN-D(ML).





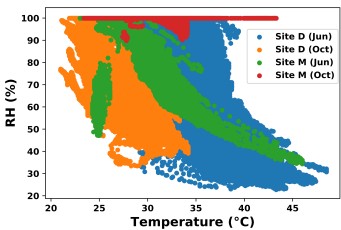

**Figure 4.** A scatter plot showing variations in RH and T at the two sites across the two deployments. The sites offer substantially diverse weather conditions. Site D exhibits wide variations in RH and T levels during both deployments. Site M exhibits almost uniformly high RH levels during the Oct deployment which coincided with the retreating monsoons.

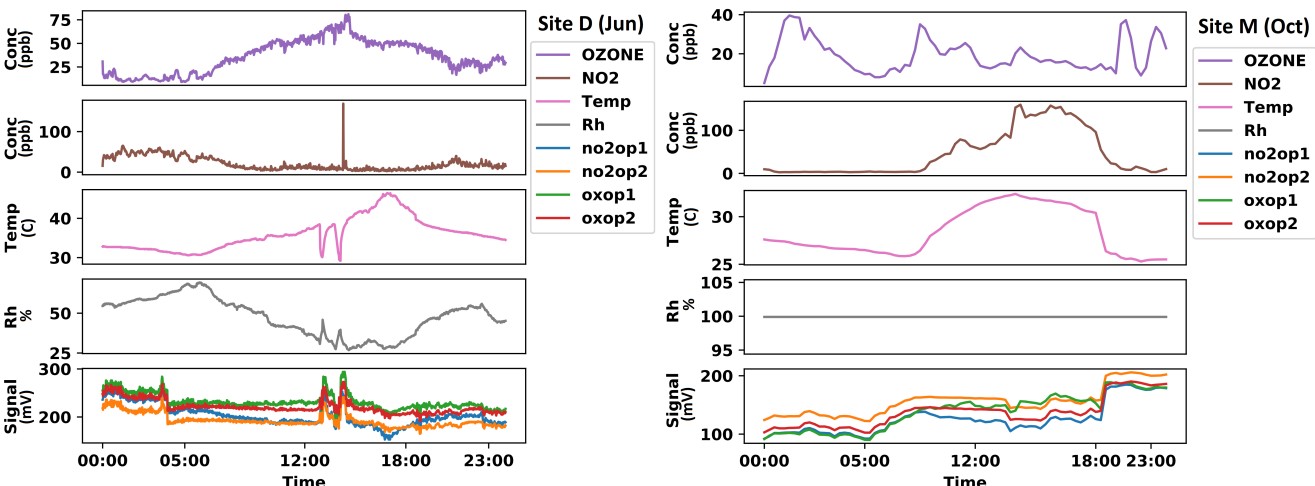

**Figure 5.** Time series showing the variation in the raw parameters measured using the reference monitors ($NO_2$ and $O_3$ concentrations) as well as those measured using the SATVAM LCAQ sensors (RH, T, no2op1, no2op2, oxop1, oxop2). The left figure considers a 24 hour periods during the Jun deployment (28 June 2019) at site D whereas the right figure considers the Oct deployment (12 October 2019) at site M. Values for site D are available at 1 minute intervals while those for site M are averaged over 15-min intervals.

## 6 Results and Discussion

The goals of using low-cost AQ monitoring sensors vary widely. This study focuses on critically assessing a wide variety of calibration models and assessing the suitability of low-cost sensors for spatially dense AQ monitoring networks.

### 6.1 Analysis of Raw Data

Our deployment strategy, consisting of two sites at geographically diverse locations and experiencing varying air pollution levels, two extended deployments during months experiencing significant variations in RH and T, as well as the swap-out experiment, were aimed at covering a wide range of real-world ambient working conditions (Cross et al., 2017). As we shall





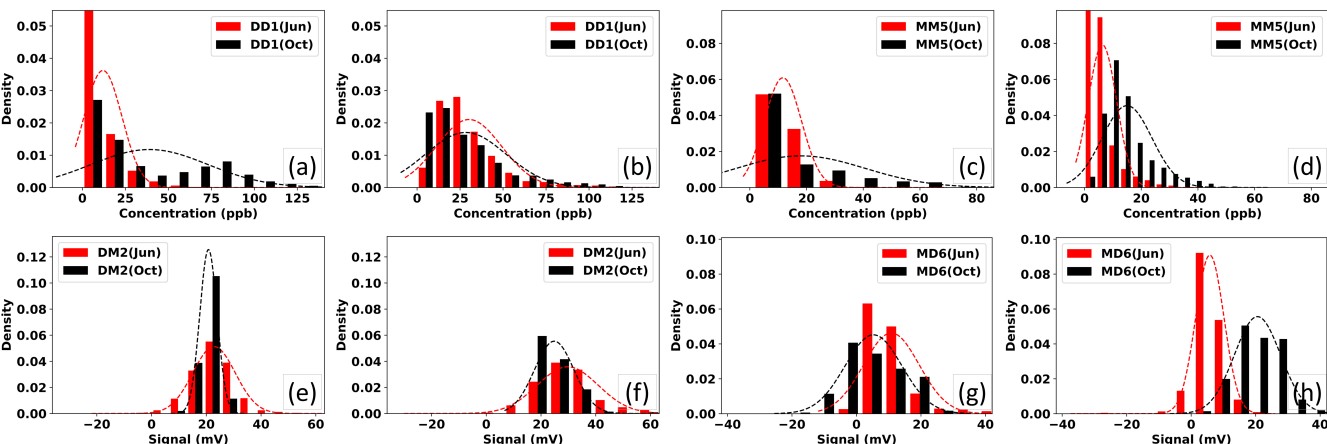

**Figure 6.** Normalized frequency distributions for various data series. Data from Jun deployments (resp Oct deployments) is shown in red (black) in all plots. The plots in row 1 show, from left to right, variations in the reference values at site D by considering data from the DD1 sensor for $NO_2$ (Fig. 6(a)) and $O_3$ (Fig. 6(b)). Fig. 6(c) and (d) show the same for site M by considering data from the MM5 sensor. Recall that both the DD1 and MM5 sensors did not participate in the swap-out experiment and remained at the same site for both deployments. The figures in row 2 plot explore cross site variations in no2diff (Fig. 6(e) and (g)) and oxdiff (Fig. 6(f) and (h)) values by considering data from the DM2 and MD6 sensors both of which participated in the swap-out experiment.

see, data from such diverse operating conditions is crucial for proper calibration of these sensors in order to not expect drastic extrapolations from the models during actual deployment.

To illustrate this, refer to Fig. 4 which shows the RH and T ranges observed during the two deployments across the two sites.
It is clear that both sites offer extremely diverse meteorological conditions, with only site M offering somewhat uniformly high RH values during the Oct deployment. We also present in Fig. 5, time series over 24 hour periods from two deployments at the two sites.

The reference data for the site D Jun deployment indicates that $O_3$ levels exhibit a diurnal trend with a midday peak mainly at around 1500 hrs, while $NO_2$ levels tend to peak usually in the morning and in the evening to midnight, suggesting nearby
roadways could be a predominant source of pollution. Site M on the other hand presents far lower $O_3$ levels. Ambient RH and T values were observed to vary inversely to each other at site D in both deployments and site M during the Jun deployment. However, site M experienced a near continuous 100% RH level during the Oct deployment. The sensor voltages (no2op1, no2op2, oxop1, oxop2) can be seen to have good correlation in the plots.

The two sites and deployments also exhibit significant diversity with respect to absolute concentrations. The reference $NO_2$
levels from site M (available at 15 minute intervals) ranged from 0.01-44.13 ppb in the Jun deployment and from 0.01-58.44 ppb in the Oct deployment, respectively. At the same time, the reference $NO_2$ levels from site D ranged from 0.70-65.49 ppb and from 0.86-159.55 ppb during the Jun and Oct deployments, respectively. Similarly, reference $O_3$ levels also differ significantly across the sites with site M levels ranging from 0.70-65.49 ppb and 0.86-160.41 ppb during the Jun and Oct deployments





respectively and those for site D ranging from 0.70-141.47 ppb and from 0.80-180.00 ppb for the same deployments. In
general, Site D experienced higher concentration levels, as well as peaks, than site M. Furthermore, concentration levels were
found to go up for both sites during the Oct deployment as compared to the Jun deployment. Such diversity in concentration
levels are expected to empower calibration models to offer accurate predictions across wide ranges of operating conditions.

As deployments experienced several cloudy days, peaks of observed $O_3$ levels are not consistent throughout the deployments.
Such influence of meteorological parameters on pollutant levels is well recognized in past literature (Gaur et al., 2014; Tiwari
et al., 2015; Simmhan et al., 2019) with effects such as scavenging of PM and gaseous pollutants that occur due to rain that may
result in lower concentration peaks of PM2.5 levels (not considered in this study) and lower mixing ratio of $NO_2$, or higher
range of concentrations of same pollutants during winter, being observed.

In order to better understand global trends in cross-site and cross-deployment variations, Fig. 6 plots histograms indicating
the statistical distribution of reference values as well as sensor voltage readings for various sites and deployments. It is notable
that both the reference values, as well as the sensor readings, seem to be statistically distributed across both sites and deploy-
ments, with the possible exception of $NO_2$ levels at site D during the Oct deployment (see Fig. 6 Row 1 left) which seems to
have a bimodal distribution.

These plots demonstrate that site D experiences appreciably greater levels for both $NO_2$ and $O_3$. This can be verified by
comparing rows 1 and 2 of Fig. 6. This is understandable since site M is located in a coastal city whereas site D is situated at
a more arid location. For both sites, in general the Oct deployment offers larger concentration levels as compared to the Jun
deployment. This is reflected in the plots in rows 3 and 4 of Fig. 6 which show that the distribution of the voltage differentials
differs significantly when the same sensor is relocated to a different site during a different season.

## 6.2   Effect of Calibration Model on Calibration Performance

We compare the calibration algorithms discussed in Sect. 4 and also those in the Supporting Information document. Given the
vast set of models that we consider, we first compare within a family of algorithms (these comparisons are presented in detail
in the Supporting Information document but summarized below as well) and present here, only comparisons across the winners
of those families. We use the Wilcoxon paired two sample test (see Sect. 3.3.2) to compare two calibration algorithms on the
same dataset. However, for visual inspection, we also provide *violin plots* of the absolute errors offered by the algorithms. See
Fig. 7 for a brief description on how to interpret a violin plot.

### 6.2.1   Interpreting the Two-sample Tests

As mentioned earlier, we used the paired Wilcoxon signed ranked test to compare two algorithms on the same dataset. Given
that there are 12 datasets and 10 splits for each dataset, for ease of comprehension, we provide globally averaged statistics of
wins scored by an algorithm over another. For example, say we wish to compare RT and KRR as done in Tab 2. We perform
the test for each individual dataset and split. For each test, we either get a win for RT (in which case RT gets a +1 score and
KRR gets 0), or a win for KRR (in which case KRR gets a +1 score and RT gets 0) or else the null hypothesis is not refuted (in





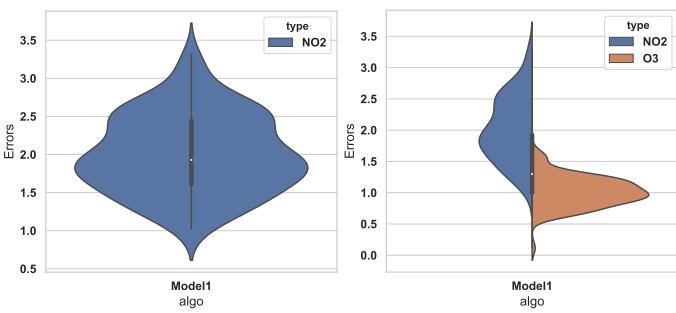

**Figure 7. (Interpreting violin plots)** Two violin plots based on synthetic error data (i.e. the data does not correspond to any actual model) are shown above. Violin plots display numeric data by showing quartile/percentile information, as well as a rotated kernel density plot to show the distribution of the data. The left figure offers a *symmetric* violin plot on a single data source ($NO_2$ calibration in this synthetic example). The thin vertical line in the middle represents the inter-percentile range between the 0.05 and 0.95 percentiles. The thicker and shorter vertical line represents the inter-quartile range between the 0.25 and 0.75 quartiles. The white dot in middle represents the median. The right figure offers a *split* violin plot that considers two data sources together for ease of comparison.

which case both get 0). The average of these scores is then shown. For example, in Tab 2 (top), row 3 column 2 records a value of 0.46 implying that in 46% of these tests, KRR won over RT in case of $O_3$ calibration, whereas row 2 column 3 records a value of 0.24 implying that in 24% of the tests, RT won over KRR. In the balance (1 - 0.46 - 0.24 = 0.3) i.e. 30% of the tests,
neither algorithm could be declared a winner.

### 6.2.2   Inter-family Comparison of Calibration Models

The calibration models described in Sect. 4 and in the Supporting Information document can be classified in to four broad families: 1) the Alphasense family (containing the four models A1 to A4), 2) linear parametric models (LS, LS(MIN) and LASSO), 3) kernel regression models (KRR and the Nystroem method), and 4) the *k*NN family including algorithms that
use metric learning (KNN, KNN-D(ML), etc) – please see the Supporting Information document for details of algorithms not described here in the main paper such as LS(MIN) etc.

A summary of the results of comparing models and algorithms *within* these families is given below. The next section will compare the winners across these families to determine a *global* winner.

1. **Alphasense**: All four Alphasense algorithms exhibit extremely poor performance across all metrics on all datasets,
offering extremely high MAE and low R2 values. This is corroborated by previous studies (Lewis and Edwards, 2016; Jiao et al., 2016; Simmhan et al., 2019).

2. **Linear Parametric**: Among the linear parametric algorithms, LS was found to offer the best performance.

3. **Kernel Regression**: We confirmed the utility of the Nystroem method as an accurate but accelerated approximation for KRR kernel ridge regression (KRR) and that the acceleration is generally higher for larger datasets.




**Table 2.** Results of the pairwise Wilcoxon signed rank tests across all model types (see Sect. 6.2.1 for a key). KNN-D(ML) beats every other algorithm comprehensively (mostly 100% of the time with the exception of NW(ML) which it still beats 58% of the time) and is scarcely ever beaten. The overall ranking of the algorithms is indicated to be KNN-D(ML) > NW(ML) > KRR > RT > LS.

| | O$_3$ | | | | | | NO$_2$ | | | | |
|---|---|---|---|---|---|---|---|---|---|---|---|
| | LS | RT | KRR | NW(ML) | KNN-D(ML) | | LS | RT | KRR | NW(ML) | KNN-D(ML) |
| LS | 0 | 0 | 0 | 0 | 0 | LS | 0 | 0.01 | 0 | 0 | 0 |
| RT | 0.98 | 0 | 0.24 | 0 | 0 | RT | 0.83 | 0 | 0.36 | 0.16 | 0 |
| KRR | 1 | 0.46 | 0 | 0 | 0 | KRR | 1 | 0.58 | 0 | 0.01 | 0 |
| NW(ML) | 1 | 1 | 1 | 0 | 0.07 | NW(ML) | 1 | 0.73 | 0.96 | 0 | 0.03 |
| KNN-D(ML) | 1 | 1 | 1 | 0.58 | 0 | KNN-D(ML) | 1 | 1 | 0.97 | 0.62 | 0 |

4. **$k$-NN and Metric Learning Models**: Among the k-NN family of algorithms, the distance weighted k-NN algorithm that uses a learnt metric i.e. KNN-D(ML) was found to offer the best accuracies across all datasets and splits.

### 6.2.3   Global Comparison of Comparison Models

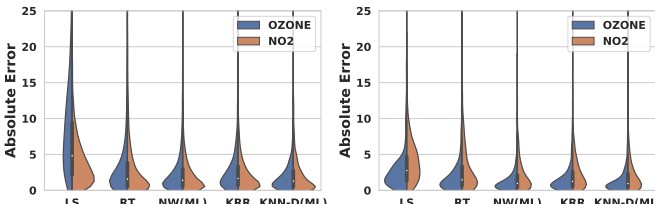

**Figure 8.** The violin plots on the left and right depict the distribution of absolute errors incurred by various models on respectively, the DD1(Oct) and MM5(Jun) datasets. KNN-D(ML) offers visibly superior performance than several other algorithms such as LS and RT.

We took the best algorithms from all families (parametric, kernel regression, $k$-NN) as well as regression trees and performed a head-to-head comparison for these to assess the winner (Alphasense models were not considered given their extremely poor performance). The KNN-D(ML) algorithm continued to emerge as the winner as indicated by the two-sample tests (Table 2) as well as the violin plots (Fig. 8).

### 6.3   The Effect of Metric Learning

Recall that in Sect. 5.1, we discussed the need for metric learning in order to place appropriate emphasis on various features, such as RH and T that are known to hugely influence calibration. To assess whether metric learning is indeed discovering such emphasis, Tab 3 shows the linear transformation corresponding to the Mahalanobis metric learnt by the NW(ML) technique for NO$_2$ calibration on the DD1(Jun) dataset. This is essentially the matrix $\Sigma^{\frac{1}{2}}$ where $\Sigma$ is the matrix corresponding to the Mahalanobis metric. We point out the following aspects of the matrix by concentrating on the diagonal entries.




**Table 3.** The linear transformation $\Sigma^{\frac{1}{2}}$ learnt for $NO_2$ calibration on the dataset DD1(Jun). Note the large emphasis the transformation places on RH and T, increasing their importance while calculating the Mahalanobis distance while placing comparatively less importance on the oxop1, oxop2 and oxdiff features which is understandable since this metric was learnt for $NO_2$ calibration.

|        | T     | RH    | no2op1 | no2op2 | oxop1 | oxop2 | no2diff | oxdiff |
|--------|-------|-------|--------|--------|-------|-------|---------|--------|
| T      | **10.19** | 3.29  | -1.95  | -2.12  | 3.73  | 4.29  | -0.66   | -1.44  |
| RH     | 3.52  | **13.22** | 1.43   | 1.46   | -2.32 | -2.60 | -0.25   | 0.49   |
| no2op1 | -0.17 | -0.69 | **6.92**   | 6.20   | -3.65 | -3.93 | 0.27    | -0.12  |
| no2op2 | -0.27 | -0.81 | 5.66   | **6.96**   | -2.94 | -3.20 | 0.51    | 0.11   |
| oxop1  | 1.27  | -0.19 | 1.94   | 2.11   | **1.51**  | 0.50  | 0.74    | 0.27   |
| oxop2  | 0.89  | -0.81 | 3.34   | 3.58   | -0.86 | **0.03**  | 0.86    | 0.24   |
| no2diff| -0.74 | -0.68 | -4.01  | -3.94  | 6.89  | 7.12  | **2.82**    | 1.88   |
| oxdiff | 2.71  | 3.45  | -7.03  | -7.36  | 7.95  | 8.54  | -0.32   | **1.32**   |

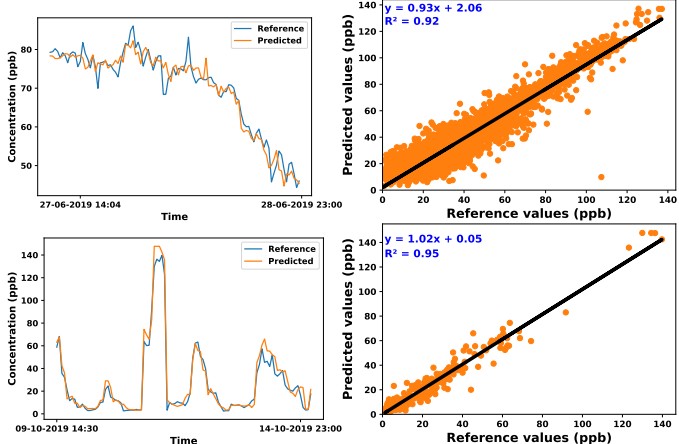

**Figure 9.** On the left hand side, the top (resp. bottom) figure exhibits a time series of the reference values and those predicted by the KNN-D(ML) algorithm for $O_3$ (resp. $NO_2$) concentrations at site D (resp. site M) during the Jun (resp. Oct) deployment. On the right hand side are scatter plots showing the correlation between the reference and predicted values of the concentrations. For both deployments, KNN-D(ML) can be seen to offer excellent calibration and agreement with the FRM-grade monitor.

1. The diagonal entries corresponding to no2op1, no2op2 and no2diff have much higher values that those for oxop1, oxop2 and oxdiff. This makes sense since this metric was being learnt for $NO_2$ calibration.

2. The diagonal entries corresponding to RH and T are by far the largest. This implies that the method did find it crucial to put more emphasis on these two features while calculating distances.

Fig. 9 presents two cases where the models offered by metric learning offer excellent agreement with the reference monitors across significant spans of time.

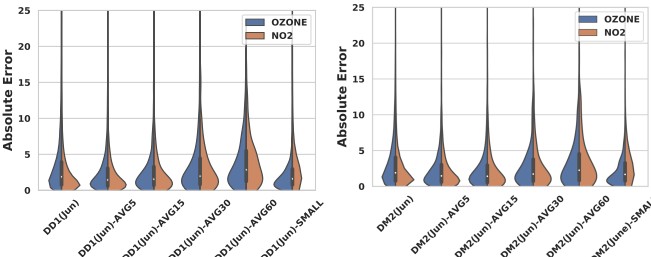

**Figure 10.** Effect of temporal data averaging, and lack of data on the calibration performance of the KNN-D(ML) algorithm on temporally averaged and sub-sampled versions of the DD1(Jun) and DM2(Jun) datasets. Notice the visible deterioration in the performance of the algorithm when aggressive temporal averaging, e.g. across 30 minute windows, is performed. $NO_2$ calibration performance seems to be impacted more adversely by lack of enough training data or aggressive averaging than $O_3$ calibration.

**Table 4.** Results of the pairwise Mann-Whitney $U$ tests on the performance of KNN-D(ML) across temporally averaged versions of the DD1 dataset (see Sect. 6.2.1 for a key). The dataset names have been abbreviated here. For example, DD1(Jun)-AVG5 is referred to as simply AVG5. These results are reported over a single split. The performance of KNN-D(ML) on AVG5 wins over its performance with any other level of averaging. It is clear that mild temporal averaging (e.g. over 5 minute windows) positively impacts calibration performance. On the other hand, the performance with extremely aggressive averaging e.g. on AVG60 is almost always inferior than any other level of averaging.

| | $O_3$ | | | | | | $NO_2$ | | | | |
|---|---|---|---|---|---|---|---|---|---|---|---|
| | DD1(Jun) | AVG5 | AVG15 | AVG30 | AVG60 | | DD1(Jun) | AVG5 | AVG15 | AVG30 | AVG60 |
| DD1(Jun) | 0 | 0 | 0 | 0 | 0 | DD1(Jun) | 0 | 0 | 0 | 1 | 1 |
| AVG5 | 1 | 0 | 1 | 1 | 1 | AVG5 | 1 | 0 | 1 | 1 | 1 |
| AVG15 | 1 | 0 | 0 | 1 | 1 | AVG15 | 0 | 0 | 0 | 1 | 1 |
| AVG30 | 1 | 0 | 0 | 0 | 1 | AVG30 | 0 | 0 | 0 | 0 | 1 |
| AVG60 | 0 | 0 | 0 | 0 | 0 | AVG60 | 0 | 0 | 0 | 0 | 0 |

## 6.4 Effect of Data Preparation on Calibration Performance

We now present studies critically assess the robustness of these calibration models, as well as identify the effect of other factors, such as temporal averaging of raw data, total amount of data available for training, and diversity in training data. We note that some of these studies were made possible only because the experimental setup enabled us to have access to sensors that did not change their deployment sites, as well as those that did change their deployment site due to the swap-out experiment.

## 6.5 Some Observations on Original Datasets

Before we proceed to perform studies with the temporally averaged, sub-sampled and aggregated datasets (see!3.3), first we look at the performance of KNN-D(ML) on the original datasets to gain some indications on the effects of these data preparation methods on calibration performance. We will then confirm these indications using the new datasets. If we consider only datasets





obtained from site D deployments, then we find that on these datasets (irrespective of whether in Jun or Oct), KNN-D(ML)
offers an extremely high average R2 of 0.952 for O$_3$ calibration. However, the same value for site M deployments (yet again
across Jun and Oct deployments) is much lower at 0.762. We observe a similar but less stark difference for NO$_2$ calibration
with site D deployments enjoying an average R2 of 0.915 with KNN-D(ML) whereas site M having only 0.846. This indicates
that paucity of data and aggressive temporal averaging may be affecting calibration performance negatively, and more directly
than seasonal variations. The above observations also indicate that O$_3$ calibration might be more sensitive to these factors than
NO$_2$ calibration.

**6.6 Effect of Temporal Data Averaging**

Recall that data from sensors deployed at site M had to be averaged over 15 minute intervals to align them with the reference
monitor timestamps. To see what effect such averaging has on calibration performance, we use the temporally averaged datasets
(see Sect. 3.3). Fig. 10 presents the results of applying the KNN-D(ML) algorithm on data that is not averaged at all (i.e. 1
minute interval timestamps), as well as data that is averaged at 5, 15, 30 and 60 minute intervals. The performance for 30 and
60 minute averaged datasets is visibly inferior that that for the non-averaged dataset. This leads us to conclude that excessive
averaging can erode the diversity of data and hamper effective calibration. To distinguish among the other temporally averaged
datasets for which visual inspection is not satisfactory, we also performed the unpaired Mann-Whitney $U$ test, the results
for which are shown in Tab 4. The results are striking in that they reveal that moderate averaging, for example at 5 minute
intervals, seems to benefit calibration performance. However, this benefit is quickly lost if the averaging window is increased
much further at which point, performance is invariably hurt.

**6.7 Effect of Data Paucity**

Since temporal averaging also decreases the amount of data as a side-effect, in order to tease these two effects apart, we also
considered the sub-sampled versions of these datasets (see Sect. 3.3). Fig. 10 also shows that reducing the amount of training
data has an appreciable negative impact on calibration performance.

**6.8 Effect of Data Volume and Diversity**

Tab 5 describes an experiment wherein we took the KNN-D(ML) model trained on one dataset and used it to make predictions
on another dataset. To avoid bringing in too many variables, this was done only in cases where both datasets belonged to the
same sensor but for different deployments. Without exception, such *transfers* led to poor performance. We confirmed that this
was true not just for calibration models learnt using non-parametric methods such as KNN-D(ML) but also parametric models
like LS or LASSO or RT.

This finding, although concerning at first, seems reasonable when we observe Fig. 4. Not only do the sites and deployments
individually span wide ranges of RH and T, but these ranges are not entirely overlapping either. Given our earlier confirmation



**Table 5.** A demonstration of the impact of data diversity and data volume on calibration performance. The first two rows present the performance of the learnt KNN-D(ML) calibration models when tested on data for a different season (deployment) but in the same site. This was done for the DD1 and MM5 sensors that did not participate in the swap-out experiment. The next two rows present the same, but for sensors DM2 and MD6 that did participate in the swap-out experiment and thus, their performance is being tested not only for a different season, but also a different city. The next four rows present the dramatic improvement in calibration performance once datasets are aggregated for these four sensors. Also notable is the fact that $O_3$ calibration seems worse affected by these variations (average R2 in first four rows being -3.68) than $NO_2$ calibration (average R2 in first four rows being -2.92).

| Train → Test | $O_3$ | | $NO_2$ | |
| --- | --- | --- | --- | --- |
| | MAE | R2 | MAE | R2 |
| DD1(Jun) → (Oct) | 28.9±1.7 | -1.63±0.39 | 33.1±0.94 | -0.87±0.08 |
| MM5(Oct) → (Jun) | 8.9±1.69 | -4.14±2.4 | 15.9±2.9 | -9.6±2.9 |
| DM2(Jun) → (Oct) | 19.0±1.3 | -8.12±1.5 | 17.1±0.97 | -0.45±0.12 |
| MD6(Jun) → (Oct) | 18.8±0.57 | -0.83±0.08 | 29.6±0.85 | -0.77±0.09 |
| DD1(Jun-Oct) | 3.3±0.14 | 0.939±0.006 | 2.7±0.06 | 0.958±0.003 |
| MM5(Jun-Oct) | 1.8±0.13 | 0.814±0.05 | 2.5±0.19 | 0.902±0.04 |
| DM2(Jun-Oct) | 3.7±0.13 | 0.909±0.009 | 3.0±0.02 | 0.762±0.008 |
| MD6(Jun-Oct) | 1.8±0.007 | 0.975±0.002 | 1.9±0.02 | 0.989±0.0006 |

of the importance these parameters have in calibration, it is not surprising that the models performed poorly when faced with unseen RH and T ranges.

To verify that this is indeed the case, we ran the KNN-D(ML) algorithm on the aggregated datasets (see Sect. 3.3) which combine training sets from the two deployments of these sensors. Tab 5 confirms that once trained on these more diverse datasets, the algorithms resume offering excellent calibration performance on the entire (broadened) range of RH and T values.

# 7    Conclusions and Future Work

In this study we presented results of a diverse field deployment of low-cost AQ monitoring sensors across two sites having
vastly different geographical, meteorological, and air pollution parameters, as well as two deployments set in seasons offering diverse RH and temperature conditions. A unique feature of our deployment was the *swap-out* experiment wherein four of the seven sensors were transported across sites in the two deployments. To perform highly accurate calibration of these sensors, we experimented with a wide variety of algorithms based on standard statistical estimation techniques but found a novel method based on *metric learning* to offer the strongest results (as verified by statistical two-sample tests) across sites and deployment
conditions at predicting both $NO_2$ and $O_3$ concentrations.

A few key takeaways from our statistical analyses are:



1. Incorporating ambient RH and T into the calibration model offers a definite advantage in achieving superior calibration performance. The inclusion of the *augmented* features oxdiff and noxdiff we describe in Sect. 3 also positively impact the performance.

2. Local methods such as $k$-NN offer the best performance on these calibration tasks. However, they stand to gain significantly through the use of metric learning techniques, which automatically learn the relative importance of each feature, as well as *hyper-local* variations such as distance-weighted $k$-NN. These indicate that these calibration tasks operate in high variability conditions where local methods offer the best chance at capturing subtle trends.

3. Performing smoothing over raw time series data obtained from the sensors may help improve calibration performance
but only if this smoothing is non-aggressive e.g. done over short windows. Very aggressive smoothing done over long windows is detrimental to calibration performance.

4. Calibration models are data-hungry as well as diversity hungry. This is especially true of local methods like $k$-NN variants. Offering these techniques limited amounts of data or even data that is limited in diversity of RH, T or concentration levels, may result in calibration models that generalize very poorly.

5. $O_3$ calibration seems to be more sensitive to unseen variations in operating conditions than $NO_2$ calibration.

Our results offer encouraging options for using low-cost AQ sensors to complement CAAQMS in creating dense and portable monitoring networks which can enable a range of studies in AQ, source apportionment, human health impacts and atmospheric chemistry studies. Among avenues for future work, an especially interesting one is the study of long-term stability of electrochemical sensors and characterizing drift or deterioration patterns in these sensors and correcting for the same. Another
interesting challenge is ultra rapid calibration of these sensors that requires minimal collocation with a reference monitor.

*Code and data availability.* The code and data used in this study are available upon request to author Purushottam Kar (purushot@cse.iitk.ac.in).

*Competing interests.* Author Ronak Sutaria is the CEO of Respirer Living Sciences Pvt. Ltd. which builds and deploys low-cost sensor based air quality monitors with trade-name 'Atmos - Realtime Air Quality' monitoring sensor networks. Ronak Sutaria's involvement was primarily in the development of the air quality sensor monitors and the big data enabled application programming interfaces to access the
temporal data but not in the data analysis. Author Brijesh Mishra, subsequent to the work presented in this paper, has joined the Respirer Living Sciences team. The authors declare no other competing interests.

*Acknowledgements.* This research has been supported under the Research Initiative for Real-time River Water and Air Quality Monitoring program funded by the Department of Science and Technology, Government of India, and Intel® and administered by the Indo-United States Science and Technology Forum (IUSSTF).





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
