# Peer review of "Robust statistical calibration and characterization of portable low-cost air quality monitoring sensors to quantify real-time $O_3$ and $NO_2$ concentrations in diverse environments"

_Atmospheric Measurement Techniques, 2020_

## Referee Comment (RC1) · Anonymous Referee #1 · 16 Jul 2020

This manuscript describes calibration of low-cost electrochemical sensors for O3 and NO2. The sensors were co-located with reference monitors in Delhi and Mumbai. A novel part of this study was that some of the sensors were swapped between cities.

Overall, it is clear that the authors have put in a huge effort to build and test various calibration models. The effort to try different models seems truly comprehensive. However, they also seem to have tried to put everything into a single paper, and as a result it is very hard for me to follow the manuscript and to see how the conclusions follow from the evidence presented in the manuscript.

[Figure]

A revised manuscript needs to be much more focused. The current version tries to cover (1)different calibration algorithms (and somehow manages to both overwhelm the reader with lots of algorithmic detail but still put much - perhaps too much - information on some algorithms in the SI), (2) the swap experiments (which get lost in the discussion of the results), (3) the value of including T and RH in the calibrations, and (4) issues of data paucity and variability. In the end it is just too much to cover in one manuscript. The authors should think critically about the central message and try to communicate that clearly.

Specific comments: Abstract line 15 - it is unclear if the $R^2$ improvement of 9% means 0.09 (e.g., 9 percentage points) or that the new $R^2$ is 1.09 times the old $R^2$. Also, if this is an important finding, it should be obvious in the manuscript text. I cannot point to the figure or table that directly supports this improvement.

Section 2.4 does not seem like it needs to be its own section. This information can be added to the end of section 2.3.

Line 203 - why are reference O3 concentrations <1 ppb scrubbed from the dataset? It seems entirely possible to have near-zero ozone in urban areas at night.

Section 3.3.1 describes splitting the data into 70% for training and 30% for testing. How is this different than the sub-sampled 2500 data points from section 3.3, item 2 in the list? (Also, it is not clear in section 6 when the 70:30 split was used versus the 2500 point subset.) The 70:30 split was repeated 10 times. I have seen 10-fold validation with a 90:10 split, which means that each data point appears in a testing dataset once. In your application of a ten-fold 70:30 split, it seems like there could be an uneven distribution of how often data show up in the training and testing subsets. Please explain why you used a 70:30 split with a 10-fold validation and give some context why this is a good approach.

Section 4 notes that "Our study used a large number of both parametric, and non-parametric calibration techniques." Those should be enumerated and briefly explained

in the text, rather than having readers hunt through the SI.

Section 5 proposes using k-NN without much explanation. Was some process used to select k-NN as the best algorithm or the one worth highlighting? It might be possible to move all of section 5 to the SI. I think the manuscript would benefit from getting to discussion of the results more quickly.

Figure 5 shows O3 increasing at site M in the middle of the night. This seems unrealistic at a time when the sun was not shining. The authors should explain what is happening or select a different example period to highlight.

The individual panels of Fig 6 should be labeled with the quantity shown. E.g., 6a should indicate NO2 on the figure panel, rather than having to dig through the caption.

Line 404 - what are RT and KRR? Those terms have not been introduced yet. General comment - this paper needs a glossary. It is very hard to keep up with all of the abbreviations.

I don't think that Figure 7 is appropriate for the main text. It should go into the SI.

Section 4 referred readers to the SI for details on the various calibration models. Now the models are described in section 6.2.2. This seems inefficient. Furthermore, the list of models should be comprehensive. Table 2 and Fig 8 both mention NW models, which of course are not listed in 6.2.2 and I had to go back 100 lines in the manuscript to find them!

General comment on length: The real meat of the paper starts in section 6 around line 425. However at this point I am pretty lost about the goals of the paper and what models are being compared. The paper is just much too long. I think most readers will quit before they get to the results.

I do not understand what Table 3 is telling me. Does interpreting this table require a good understanding of the Mahalanobis metric?

Figure 9 - please label the subpanels (a), (b), (c), (d) and explain each panel in the caption.

All of the violins in Fig 8 and 10 have instances of very high error. Are these occasions the same for all tested models? And what conditions lead to high error? I think that AMT readers will be more interested in model performance - e.g., periods when the models perform poorly - than the depth of mathematical detail presented e.g., in sections 4 and 5.

Section 6.4 seems like it should either be removed or significantly expanded. Right now it does not seem to add anything to the manuscript.

One of the main aspects of this study, at least as explained in the Intro, was the sensor swap between sites D and M. As far as I can tell, this was not discussed at all in the Results and Discussion. What happened to sensors moved from D to M and vice versa? Performance of calibration models built in city A and applied in city B is of major interest to the low-cost sensor community, especially as more sensors get deployed in locations that currently lack reference monitors needed to build local calibration models.

Minor and grammar Line 94 says that the sensors "include contain". Please edit.

Line 118 - does the Plantower output milligram/mˆ3 or microgram/mˆ3?

Line 133 - sites D and M are undefined.

---

## Referee Comment (RC2) · Anonymous Referee #3 · 31 Jul 2020

General Comments:

This paper provides a comprehensive investigation of several factors affecting the calibration of low-cost sensors across different seasons and environments. While some aspects of the presentations of the results could be improved to enhance clarity, the basic approach appears sound. Most of my suggestions are related to specific changes that should be made, as outlined under the "specific comments" section below. In addition to these, I have some general comments, which should be taken more as suggestions of possible topics for further investigation or discussion rather than recom-

mendations of anything that must be improved in this paper.

It was mentioned in Section 3.2 that 52% of timestamps were invalid. Does this mean that more than half of the collected data was corrupted in some way, or simply that not all sensor were operating at all times? If the former, this would be concerning, as it means that less than half of the data during a long-term deployment might be useable, which would throw into question the validity of, for example, long-term average measurements. While your investigation of the effects of "small" datasets indicates this might not be too detrimental to overall performance, if such missing data were clustered in time rather than truly random, this might introduce biases into the dataset.

With respect to the "Aggregated Datasets", while it is possible to identify the effects of seasonally diverse datasets and of seasonally and spatially diverse data, investigating the effects of spatially diverse data only is not possible with the current setup (since the datasets which are spatially diverse feature data from different seasons as well). Have you investigated the creation of a dataset which is diverse in terms of the location but not in season (e.g. combining DD1(Jun) and MM5(Jun))? Of course, this would introduce the effects of sensor-to-sensor variability into the results, but that could also be investigated by comparing how, for example, a calibration on DD1 could apply to DD3. While you observe that parametric and non-parametric methods both suffer from poorer performance when generalizing beyond their calibration site and season, it would be interesting to see if parametric and non-parametric methods suffered to different degrees. Based on my experience, parametric methods may perform more poorly that non-parametric methods on data drawn from the same distribution as the training data, but generalize better than non-parametric methods when the underlying distribution in changed, i.e. by moving to a new site or climate.

Specific Comments:

Line 4: I don't know what "commodity" refers to here.

Lines 4-5: I would rather say that low-cost sensors can possess high precision (i.e.

high consistency between measurements), but require calibration to attain accuracy (i.e. similarity of low-cost measurements to those of regulatory instruments).

Lines 14-15: The "2" in "R2" is not appearing as a superscript (this seems to be true throughout the document).

Line 15: "upto" should be "up to".

Line 52: Again, I am uncertain about what "commodity" means in this context.

Line 94: The word "include" appears redundant here.

Lines 94-96: The "2.5" of "PM2.5" should be subscripted. This occurs several times throughout the document.

Line 116: Again, I am uncertain about what "commodity" means in this context.

Line 118: The "2.5" of "PM2.5" should be subscripted.

Line 146: "sensor" is repeated.

Figure 3: It is mentioned elsewhere that four sensors are swapped between the two deployment locations, yet only three are listed in the figure.

Line 182: It is mentioned that there are seven sensors, but only six are listed.

Lines 184-186: Again, only three of the mentioned four sensors are listed.

Line 188: I believe "commodity" means "commercially available" or "commercially purchased" in this context. I would assume all other references to "commodity" have similar meanings as well.

Line 196: It should be specified if the T and RH values come from the internal sensors within the LCAQ sensors, or if these are from the reference instrument.

Figure 6: This figure could be clarified significantly. First, for the upper row, the pollutant in question should be specified on each plot. Also, since these plots refer to values measured by the reference instrument, specifying the LCAQ sensor data set from which the measures are plotted is unnecessary and potentially misleading. Instead, for example, sub-figure a could be titled "NO2 at site D", and the two colors could indicate simply "Jun" or "Oct". For the second row, I would also recommend simplifying the labels and switching data around so that each figure compares the two sites for the same season; as it is currently presented, differences are shown for both season and site, which makes it harder to separate these two effects. So, for example, sub-figure e could present "no2diff in Jun" with the two colors representing sites "D" and "M". If sensor-to-sensor differences are a concern, you could use the average or median of values across all sensors deployed to a common site. However, I think an even easier approach would be to instead continue to use the reference data; since this figure is mainly serving to show how concentrations vary by site and by season, using only reference data (combined in different ways depending on what comparisons are being made) would be a valid approach. If you still want to include a comparison of the raw signal differences, I would recommend moving that to the supplemental information; I think trying to do that in one figure which properly accounts for all three sources of variability (season-to-season differences, site-to-site differences, and sensor-to-sensor differences) would be too complicated for a clear main-paper result. Finally, the dotted curves in the figures are not described. They appear to represent Gaussian distributions fit to the data, but since that is not an appropriate distribution for these data (e.g. they are strictly non-negative for the top row), I would recommend omitting these as they can be potentially misleading.

Line 385: What does "statistically distributed" mean?

Line 391: The figure does not appear to have 3rd or 4th rows.

Figure 7: I recommend moving this to the supplemental information; Violin plots are sufficiently common that they do not need to be described in such detail in the main text. However, in the case of the split violin plot, you should specify whether the depicted median and ranges refer to what is plotted on the left or right halves of the plot (or, are

two such depictions provided?).

Figure 9: Whether the figure refers to NO2 or O3 should be clearly denoted in the figures themselves, rather than just in the caption. This will minimize potential confusion and misunderstandings.

Section 6.4: This section appears incomplete. It does not refer to any specific results or figures. It is possible that sections 6.5 through 6.8 were meant to be sub-sections of this section.

Section 6.7: Although it has been mentioned elsewhere, I would recommend restating in the main body of the text that you observe that the effect is more severe for NO2 data than for O3 data.

---

## Author Comment (AC1) · 6 Sep 2020

Response to Anonymous Referee #1

============================

We thank the referee for the comments and suggestions. Below we offer some clarifications in response to the same. The revised manuscript will be communicated to the editorial desk shortly.

[Figure]

1. Focus and length of manuscript: A major revision to the manuscript has been prepared that offers a more focussed discussion. The revision reorganizes portions that significantly impact the overall discussion of characterizing calibration models. All other material e.g. Figure 7, has been relegated to the supplementary material. We thank the referee for pointing out the inefficiencies in the structuring of the paper.

2. Formatting: we thank the referee for taking pains to point out several improvements in typography and formatting e.g. merging subsections, adding a glossary, labels in Figures 6, 9 etc. We have incorporated all of them in the revised version of the manuscript.

3. Abstract: improvements are indeed in percentage points. In the revision, we have included tabular values in the supplementary material which explicitly quantify the improvements, in addition to the violin plots which offer a more visual interpretation. These values are referred to in the main text in the revised version.

4. O3 concentrations: we thank the referee for pointing this out. We indeed discarded only those reference monitor values that were less than 0ppb and have corrected this typographical error. The reference monitors sometimes offer negative readings when powering up and under some other anomalous operating conditions e.g. condensation at the inlet. However, we note that less than 0.1% of the valid timestamps had reference O3 values between 0 and 1 ppb.

5. Train-test splits: we chose a 70:30 split since it gave us sizeable sets for both training and testing. Machine learning and statistical estimation literature uses various splits such as 70:30, 80:20, etc. Our splits were repeated independently 10 times to allow two-sample tests to be carried out. The k-fold split method as mentioned by the referee, is another alternative. However, with k=10, the resulting 90:10 split offers a rather small test set which we wished to avoid. We verified that the choice of the size of the split (e.g. 70:30 vs 80:20) does not alter the conclusions of the paper.

6. Subsampled datasets: to create the subsampled datasets in section 3.3, we took a split (a split being a 70-30 division among train and test) and randomly subsampled

2500 points from the training portion of the split. The test portion was not altered since the aim of this experiment was to study how lack of training data affects calibration performance. To be more specific, if a dataset contained a total of 10000 valid timestamps, the train-test splits would contain resp. 7000-3000 points. For the subsampled version of this dataset, we would sample 2500 points from the 7000 points, train on those 2500 points and then test on the 3000 test points.

7. Motivation behind choice of k-NN: k-NN and kernel estimators (kernel ridge regression (KRR) and Nadaraya-Watson (NW)) are well studied non-parametric estimators in literature. These are also known to be asymptotically universal which theoretically guarantees their ability to accurately model complex patterns when given diverse and sufficient data.

8. Figures: we thank the referee for pointing out the improvements to the figures. We have replotted figures 5 and 9 in the revision to consistently show results across the same two full days of operation (01-02 July and 20-21 Oct) for sake of clarity. These figures in the old version chose different days as well as different durations which we agree was inefficient. Figure 5 had a manual labelling error (high O3 levels in the night) which we have corrected in the revision.

9. Table 3: A metric tells us how to compute distance between two points, say 8 dimensional vectors in our case. The Euclidean metric gives equal importance to all 8 dimensions when calculating distances. An alternative interpretation of a Mahalanobis metric is that it tells us how to reorganize dimensions/features so that the resulting distances, when used by the kNN algorithm, give better performance. Table 3 shows us the optimal reorganization found by the metric learning technique. In particular, note that it places heavy emphasis on the Rh and T features. This means that the optimal Mahalanobis metric identifies that a high importance should be placed on Rh and T features when computing distances for use by kNN.

10. Regions of high error: The (revised) supplementary material now contains an

analysis of regimes in which various algorithms offer larger errors. The (Rh, T) space was divided into various buckets to analyze the performance of each algorithm in each bucket. For data hungry non-parametric algorithms such as RT, NW(ML), KRR, and KNN-D(ML), regions of larger error coincided almost entirely with regions where data was scarce. This is as expected. The least squares method on the other hand demonstrated no such clear trend on regions of high error. We also tracked the errors of various algorithms across the day and found that for O3, whose diurnal levels are more predictable, all algorithms tended to offer relatively larger errors when the (true) O3 levels were higher (i.e. during peak sunlight hours). For NO2, which demonstrates no such predictable diurnal patterns, no patterns in errors were observed either.

11. Section 6.4: we submit that sections 6.5 through 6.8 were meant to be subsections of section 6.4. We have corrected this formatting error in the revision.

12. Swap experiment: Table 5 does discuss cases when sensors are trained in one season and tested in another season. Cases are considered when the site is kept the same across seasons, as well as when the site is changed across seasons. We also request the referee to take a look at the comment of Referee #3 on this point and our rebuttal to the comment (please see "General Comments" bullet point 2 in our response to Referee #3).

13. The Plantower PMS7003 offers readings in microgram per cubic meter. We thank the referee for pointing out the correction and have made the same in the revision.

---

## Author Comment (AC2) · 6 Sep 2020

Response to Anonymous Referee #3

============================

We thank the referee for the comments and suggestions. Below we offer clarifications in response to the general and specific comments of the referee. The paper has undergone a major revision that will be communicated to the editorial desk shortly.

General Comments
* * *
1. Invalid Timestamps: although around half the timestamps were rejected for our experiments, it was still the case, especially for summer months, that at least one timestamp (frequently several) were found valid every hour. We note that this does not contradict the rejection of 52% timestamps since site D (resp. site M) offered timestamps at 1 minute (resp. 15 minute) intervals. Thus, the timestamps considered valid could still accurately track diurnal changes in AQ parameters (as indicated by Figure 9). A conservative approach was adopted when rejecting timestamps. We recall that a total of 8 parameters are involved in the training process – four voltage values, relative humidity and temperature values from the LCAQ sensor, and two reference values (one each for O3 and NO2) from the reference monitors. Timestamps where even one of these parameters had an invalid value were rejected. Table 1 has been revised to include more illustrative examples of rejected timestamps. In future work, data imputation techniques could be adopted to increase the number of valid timestamps. We have included a discussion on this point in the (revised) supplementary material.

2. Spatially Diverse Data: the prospect of investigating the effect of spatial variation alone (without bringing seasonal variations into account) is interesting and we did consider this in our initial experiments but found that cross-sensor calibration is a challenging task in itself. For example, even the relative humidity and temperature sensors present in LCAQ sensors do not present good agreement across sensors (please also see point 4 in specific comments below). Thus, investigating spatial variation alone would have required us to do some form of "model transfer" of calibration models from one LCAQ sensor to another. Although an encouraging direction for future work, this was not within the scope of this paper.

3. Parametric vs non-parametric for out-of-sample results: we thank the referee for making this suggestion. We have updated Table 5 to include the generalization results

for the parametric linear least squares method LS as well. As noted by the referee, performance drops are noticed in both algorithmic paradigms. However, as compared to the non-parametric method KNN-D(ML), the drop for LS is less in some cases, but comparable or worse in others. Of course, when diverse data is provided to both algorithms, KNN-D(ML) is superior at exploiting the additional diversity in data.

Specific Comments
* * *
1. We thank the referee for taking pains to point out several typographical and typesetting changes. We have incorporated all of them in the revised version of the manuscript. 2. Commodity: the Alphasense electrochemical sensors used in the SATVAM LCAQ setup were not customized or specifically tailored for our study. Hence we use the term "commodity" to describe them. We have clarified this term at its point of first use in the paper.

3. Sensor Count: we thank the referee for pointing this out. It seems we forgot to include a clarificatory remark in the paper. There were indeed 7 sensors deployed in the field of which 4 were swapped across sites. However, one of the sensors (that was swapped) was experiencing severe malfunction. Its Rh and T sensors were non-functional for the entire duration of the Jun deployment. For the Oct deployment, its data had much larger gaps (sometimes spanning several days), which was qualitatively distinct from the other sensors which experienced only intermittent gaps often lasting a few minutes. For this reason, this sensor was excluded from our study. Although for sake of full disclosure we still mentioned in our original submission that 7 sensors were used, we forgot to include this clarificatory remark. We have now included this clarification in the revision.

4. Rh and T values were obtained from DHT22 sensors located in the individual LCAQ sensors. This was done to ensure that the calibration models, once trained, could perform predictions using data available from the LCAQ sensor alone and not rely on

data from a reference monitor.

5. Figure 6: we agree with the referee and have moved this plot to the supplementary material in the revised version as the plot offers marginal utility with respect to the main discussion. We have also clarified all aspects of the plot in the caption itself as kindly pointed out in the review.

6. "Statistically distributed": we thank the referee for pointing out this typographical error. We meant to write "normally distributed". We have corrected this.

7. Number of rows in figure 6: we regret this formatting error. Our initial submission to the journal was in a two column format (in which Figure 6 did have 4 rows). However, we were requested by the editorial desk to convert to a single column format. We did so but forgot to change this piece of text to reflect the change in formatting. We have corrected this.

8. We agree and have moved the small tutorial on interpreting violin plots to the supplementary.

9. Section 6.4: the referee is indeed correct in observing that sections 6.5 through 6.8 were meant to be subsections of section 6.4 We have corrected this.

---

## Author Response (AR1)

**Cover Letter and Author Responses**

**Point-by-point Responses to Anonymous Referee #1**

1. **Query/Comment**: *A revised manuscript needs to be much more focused.*
   **Response**: We thank the referee for this suggestion. A major revision to the manuscript has been prepared that offers a more focussed discussion.
   **Edits to Manuscript**: The revision tightens the presentation, and reorganizes portions to retain only those that impact the overall discussion of characterizing calibration models. Other portions e.g. Figure 7, has been moved to the supplementary material. We thank the referee for pointing out the inefficiencies in the structuring of the paper.

2. **Query/Comment**: *Abstract line 15 (performance improvements and missing reference)*
   **Response**: improvements are indeed in percentage points.
   **Edits to Manuscript**: In the revision, we have included numerical values in a separate table (a new Table 4) in the main paper which explicitly quantify the improvements. These numerical values are referred to in the abstract and the main text to establish the claimed improvement. The text in section 6.5 (5.2.1 in the revision) has also been modified to similarly refer to Table 4 values to substantiate its claims.

3. **Query/Comment**: *Line 133 - sites D and M are undefined*
   **Response**: we thank the referee for pointing this missing reference.
   **Edits to Manuscript**: We have reordered the subsections in the revision so that sites D and M are defined before they are referred to in the discussion.

4. **Query/Comment**: *Section 2.4 does not seem like it needs to be its own section.*
   **Response**: We agree.
   **Edits to Manuscript**: The revised manuscript merges this section within section 2.3.

5. **Query/Comment**: *Line 203 - why are reference O3 concentrations <1 ppb scrubbed from the dataset?*
   **Response**: We thank the referee for pointing this out. We indeed discarded only those reference monitor values that were less than 0ppb and have corrected this typographical error. The reference monitors sometimes offer negative readings when powering up and under some other anomalous operating conditions e.g. condensation at the inlet. However, we note that less than 0.1% of the valid timestamps had reference O3 values between 0 and 1 ppb.
   **Edits to Manuscript**: The typographical error has been corrected.

6. **Query/Comment**: *Creation of train-test splits in 70-30 ratio in section 3.3.1*
   **Response**: we chose a 70:30 split since it gave us sizable sets for both training and testing. Machine learning and statistical estimation literature uses various splits such as 70:30, 80:20, etc. Our splits were repeated independently 10 times to allow two-sample tests to be carried out. The k-fold split method as mentioned by the referee, is another alternative. However, with k=10, the resulting 90:10 split offers a rather small test set which we wished to avoid. We verified that the choice of the size of the split (e.g. 70:30 vs 80:20) does not alter the conclusions of the paper.

7. **Query/Comment**: *Creation of subsampled datasets in section 3.3*
   **Response**: To create the subsampled datasets in section 3.3, we took a split (a split being a 70-30 division among train and test) and randomly subsampled 2500 points from the training portion of the split. The test portion was not altered since the aim of this experiment was to study how lack of training data affects calibration performance. To be more specific, if a dataset contained a total of 10000 valid timestamps, the train-test splits would contain resp. 7000-3000 points. For the subsampled version of this dataset, we would sample 2500 points from the 7000 points, train on those 2500 points and then test on the 3000 test points.
   **Edits to Manuscript**: We have clarified this in the paper and reordered the placement of section 3.3.1 to better explain this process.

8. **Query/Comment**: *Section 4: enumeration of techniques studied in the paper*
   **Response**: We agree with the referee and are thankful for the suggestion.
   **Edits to Manuscript**: We have included a glossary enumerating the names and brief descriptions of all techniques so that references to various algorithms can be readily checked. In order to improve the focus of the paper as suggested by the referee, we have moved details of other algorithms to the supplementary material.

9. **Query/Comment**: *Reason behind choosing k-NN style algorithms*
   **Response**: k-NN and kernel estimators (kernel ridge regression (KRR) and Nadaraya-Watson (NW)) are well studied non-parametric estimators in literature. These are also known to be asymptotically universal which theoretically guarantees their ability to accurately model complex patterns when given diverse and sufficient data.
   **Edits to Manuscript**: A brief explanation has been included in section 5.

10. **Query/Comment**: *Figure 5 - replotting*
    **Response**: we thank the referee for pointing out the improvements to the figures. Figures 5 and 9 in the old version chose to show data for different days as well as different durations which was inefficient. Figure 5 also had a manual labelling error which we have corrected in the revision.
    **Edits to Manuscript**: We have replotted figures 5 and 9 in the revision to consistently show results across the same two full days of operation (01-02 July and 20-21 Oct) for sake of clarity. However, figure 5 has been moved to the supplementary to make the discussion more focused.

11. **Query/Comment**: *Interpreting Table 3*
    **Response**: A metric tells us how to compute distance between two points, say 8 dimensional vectors in our case. The Euclidean metric gives equal importance to all 8 dimensions when calculating distances. An alternative interpretation of a Mahalanobis metric is that it tells us how to reorganize dimensions/features so that the resulting distances, when used by the kNN algorithm, give better performance. Table 3 shows us the optimal reorganization found by the metric learning technique. In particular, note that it places heavy emphasis on the Rh and T features. This means that the optimal Mahalanobis metric identifies that a high importance should be placed on Rh and T features when computing distances for use by kNN.
    **Edits to Manuscript**: A clarification has been added to the revision. However, we felt

that this discussion was not key to the focus of the paper and have moved this discussion to the supplementary material.

12. **Query/Comment**: *Analyzing where various algorithms offer high error in Figures 8, 10*
**Response**: We thank the referee for this suggestion. To do this analysis, the (Rh, T) space was divided into various buckets to analyze the performance of each algorithm in each bucket. For data hungry non-parametric algorithms such as RT, NW(ML), KRR, and KNN-D(ML), regions of larger error coincided almost entirely with regions where data was scarce. This is as expected. The least squares method on the other hand demonstrated no such clear trend on regions of high error. We also tracked the errors of various algorithms across the day and found that for O3, whose diurnal levels are more predictable, all algorithms tended to offer relatively larger errors when the (true) O3 levels were higher (i.e. during peak sunlight hours). For NO2, which demonstrates no such predictable diurnal patterns, no patterns in errors were observed either.
**Edits to Manuscript**: The revision now contains a new figure 6 and an analysis of situations in which various algorithms offer larger errors.

13. **Query/Comment**: *Section 6.4 typesettting*
**Response**: Sections 6.5 through 6.8 were meant to be subsections of section 6.4.
**Edits to Manuscript**: We have corrected this formatting error in the revision.

14. **Query/Comment**: *Include a discussion on swapout experiments where training and prediction are done across seasons or sites.*
**Response**: Table 5 and section 6.8 in the old version do discuss cases when sensors are trained in one season and tested in another season, which does include cases when the site changes across seasons. We also request the referee to take a look at the comment of Referee #3 on this point and our rebuttal to the comment (please see "General Comments" bullet point 2 in our response to Referee #3).
**Edits to Manuscript**: We have changed the title of subsection (section 5.2.4 in the revision) discussing the swapout experiment to highlight this.

15. **Query/Comment**: *Line 118 - Plantower output*
**Response**: The Plantower PMS7003 offers readings in microgram per cubic meter, We thank the referee for pointing this out.
**Edits to Manuscript**: We have corrected the typographical error in the revision.

16. **Query/Comment**: *Various comments on typography and formatting (e.g. line 133 - defining sites D and M, line 94, adding a glossary, labelling panels in figures 6 and 9, moving figure 7 to supplementary material)*
**Response**: We thank the referee for taking pains to point out several improvements in typography and formatting.
**Edits to Manuscript**: All suggestions have been incorporated in the revision.

**Point-by-point Responses to Anonymous Referee #3**

**General Comments**

1. **Query/Comment**: *Invalid timestamps: were 52% datapoints indeed discarded?*
**Response**: Although around half the timestamps were indeed rejected (those that had

even one invalid measurement), it was still the case, especially for summer months, that at least one timestamp (frequently several) were found valid every hour. We note that this does not contradict the rejection of 52% timestamps since site D (resp. site M) offered timestamps at 1 minute (resp. 15 minute) intervals. Thus, the timestamps considered valid could still accurately track diurnal changes in AQ parameters (as indicated by Figure 9). A conservative approach was adopted when rejecting timestamps. We recall that a total of 8 parameters are involved in the training process -- four voltage values, relative humidity and temperature values from the LCAQ sensor, and two reference values (one each for O3 and NO2) from the reference monitors. Timestamps where even one of these parameters had an invalid value were rejected. In future work, data imputation techniques could be adopted to increase the number of valid timestamps.

**Edits to Manuscript**: We have included a discussion on this in the revision. Table 1 has been revised to include more illustrative examples of rejected timestamps.

2. **Query/Comment**: *Creation of a dataset that is diverse w.r.t. location but not season*
**Response**: The prospect of investigating the effect of spatial variation alone (without bringing seasonal variations into account) is interesting and we did consider this in our initial experiments but found that cross-sensor calibration is a challenging task in itself. For example, even the relative humidity and temperature sensors present in LCAQ sensors do not present good agreement across sensors. Thus, investigating spatial variation alone would have required us to do some form of "model transfer" of calibration models across LCAQ sensors. This is an encouraging direction for future work.
**Edits to Manuscript**: We have added a short discussion about this in section 3.2 itself where the derived datasets are discussed.

3. **Query/Comment**: *Out-of-sample generalization of parametric vs non-parametric models*
**Response**: we thank the referee for making this suggestion. As noted by the referee, performance drops are noticed in both algorithmic paradigms. However, as compared to the non-parametric method KNN-D(ML), the drop for LS is less in some cases, but comparable or worse in others. Of course, when diverse data is provided to both algorithms, KNN-D(ML) is superior at exploiting the additional diversity in data.
**Edits to Manuscript**: We have updated Table 5 (table 6 in the revision) to include the generalization results for the parametric linear least squares method LS as well.

**Specific Comments**

1. **Query/Comment**: *Various comments on typography and formatting (e.g. "upto" vs "up to", typesetting 2.5 as a subscript in PM2.5, typesetting 2 as a superscript in R2, gas labels in figure 9)*
**Response**: We thank the referee for taking pains to point out several typography and typesetting changes.
**Edits to Manuscript**: We have incorporated all changes in the revised version.

2. **Query/Comment**: *Abstract: LCAQ are consistent but require calibration for accuracy.*
**Response**: We thank the referee for suggesting this rewording and agree with the same.
**Edits to Manuscript**: We have incorporated all changes in the revised version.

3. **Query/Comment**: *Lines 4, 52, 116, 188: reference to the word "commodity"*
   **Response**: The Alphasense electrochemical sensors used in the SATVAM LCAQ setup were not customized or specifically tailored for our study. Hence we use the term "commodity" to describe them.
   **Edits to Manuscript**: We have clarified this term at its point of first use in the paper.
4. **Query/Comment**: *Figure 3, lines 182, 184-186: number of sensors getting swapped*
   **Response**: We thank the referee for pointing this out. It seems we forgot to include a clarificatory remark in the paper. There were indeed 7 sensors deployed in the field of which 4 were swapped across sites. However, one of the sensors DM4 (that was swapped) was experiencing sensor malfunction. Its onboard Rh and T sensors were non-functional for the entire duration of the Jun deployment. For the Oct deployment, its data had extremely large gaps (sometimes spanning several days), which was qualitatively distinct from the other sensors which mostly experienced only intermittent gaps lasting a few minutes. For this reason, this sensor was excluded from our study. Although for sake of full disclosure we still mentioned in our original submission that 7 sensors were used, we forgot to include this clarificatory remark.
   **Edits to Manuscript**: We have included this clarification in the revision and corrected the number of sensors reported at various places in the paper to be consistent.
4. **Query/Comment**: *Do Rh, T values come from: LCAQ sensors or reference monitors?*
   **Response**: Rh and T values were obtained from DHT22 sensors located in the individual LCAQ sensors. This was done to ensure that the calibration models, once trained, could perform predictions using data available from the LCAQ sensor alone and not rely on data from a reference monitor.
   **Edits to Manuscript**: We have clarified this in the revision in section 3.
5. **Query/Comment**: *Clarify figure 6 labels, add plots showing site variation, and avoid Gaussian fitting for unsigned data*
   **Response**: we agree with the referee's suggestions and are thankful for the same.
   **Edits to Manuscript**: We have moved this plot to the supplementary material in the revised version as well as added plots that show differences across sites but in the same season. We have also clarified all aspects of the plot as kindly pointed out in the comments. We have also replaced Gaussian fits (dotted lines) with non-parametric KDE fits which are more appropriate for data that is visibly non-Gaussian.
6. **Query/Comment**: *Line 385: What does "statistically distributed" mean?*
   **Response**: We thank the referee for pointing out this typographical error. We meant to write "normally distributed". However, we have amended this statement since some of the distributions do not seem normally distributed.
   **Edits to Manuscript**:  We have corrected this typographical error in the revision.
7. **Query/Comment**: *Line 391: The figure does not appear to have 3rd or 4th rows.*
   **Response**: We regret this formatting error. Our initial submission to the journal was in a two column format (in which Figure 6 did have 4 rows). However, we were requested by the editorial desk to convert to a single column format. We did so but forgot to change this piece of text to reflect the change in formatting. We have corrected this.
   **Edits to Manuscript**:  We have corrected this formatting error in the revision. However, the figure and accompanying discussion has been moved to the supplementary material.

8. **Query/Comment**: *shifting figure 7 to the supplementary and clarifying violin plot details*
   **Response**: We agree. We used the standard Python-based library seaborn to create the plots. Seaborn seems to calculate medians and interquartile range of the combined left and right data in the case of split violin plots. This can be seen in figure 7 (right) where the median and interquartile ranges correspond to the combined data rather than the left or the right data.
   **Edits to Manuscript**: We have moved the small tutorial on interpreting violin plots to the supplementary and added this clarification on medians and interquartile ranges.
9. **Query/Comment**: *Section 6.4 appears incomplete*
   **Response**: The referee is indeed correct in observing that sections 6.5 through 6.8 were meant to be subsections of section 6.4
   **Edits to Manuscript**: We have corrected this formatting error  in the revision.

**General Comments on Edits to the Manuscript**

Apart from changes to fix typographical or formatting errors (e.g. repeated words "sensor sensor", "inter" vs "intra" in title of section 6.2.2, labelling errors, subscript error in formatting PM2.5, R2), most changes were done to improve the focus of the paper and make the writing more concise. We agree with the comments of both referees that encouraged us to move portions not essential to the core discussion, to the supplementary material.

1. Section 6.1 (Analysis of Raw Data) has been moved to the supplementary material along with detailed descriptions of the deployment sites in section 2.2. It was suggested in the review that Figure 6 etc be moved to the supplementary and we agreed that these portions do not significantly contribute to the core discussion.
2. As suggested in the review, portions of sections 4 and 5 have been moved to the supplementary material. The main text now briefly outlines baseline calibration methods, motivates the proposed method and gives necessary details of the proposed method. We agree with the referee comments on making the presentation of the calibration algorithms tighter.
3. Additional results have been introduced in the main text as suggested in the referee comments, for which we are thankful. For example, a glossary of the acronyms used in the discussion, results of the parametric algorithm LS in the swapout experiments on the aggregated datasets, a discussion on the cases in which various algorithms offer high error, and numerical values of performance improvements offered by the proposed method, in addition to the violin plots.
4. Algorithm 1 in the main paper has been simplified to describe only the proposed KNN-D(ML) algorithm. Earlier the algorithm sought to describe the entire family of KNN-style algorithms which may have been confusing. The general description of the KNN family of algorithms that was earlier present in the main text has been moved to the supplementary material for the interested reader.
5. The discussion around the diagonal entries of the learnt metric and the accompanying Table 3 have been moved to the supplementary material. It seems that the discussion may not be of general interest.

[revised manuscript text omitted]

---

## Author Response (AR2)

**Cover Letter and Author Responses**

**Point-by-point Responses to Anonymous Referee #1**

1. **Query/Comment**: *I have a minor comment on table 3 - it would be helpful if the caption was clearer about what the numbers mean.*
   **Response**: We thank the referee for this suggestion. We have added clarifications to the captions of tables 3 and 5 with a pointer to a discussion on how to interpret the numbers in the tables.
   **Edits to Manuscript**: The clarifications have been added as requested.

**General Comments on Edits to the Manuscript**

We have made minor changes correcting grammatical errors, spelling mistakes, and improving presentation at places. Apart from this, the only major change has been shifting Figure 2 from the supplementary material to the main paper since we felt that this figure allows the reader to visualize the data better. The changes (tracked using latexdiff) are submitted along this cover letter.

[revised manuscript text omitted]